health and disease and epidemiology/theoretical biology

SARS-CoV-2, COVID-19, vaccine, vaccine escape, heterogeneous population, policy

**Author for correspondence:**
Julia R. Gog
e-mail: jrg20@cam.ac.uk

# Vaccine escape in a heterogeneous population: insights for SARS-CoV-2 from a simple model

Julia R. Gog[1,2], Edward M. Hill[2,3,4,5], Leon Danon[2,6,7] and Robin N. Thompson[2,3,4]

[1]Department of Applied Mathematics and Theoretical Physics, University of Cambridge, Cambridge, UK
[2]JUNIPER – Joint UNIversities Pandemic and Epidemiological Research, UK
[3]The Zeeman Institute for Systems Biology and Infectious Disease Epidemiology Research,
[4]Mathematics Institute, and [5]School of Life Sciences, University of Warwick, Coventry, UK
[6]Department of Engineering Mathematics, University of Bristol, Bristol, UK
[7]The Alan Turing Institute, London, UK

JRG, 0000-0003-1240-7214; EMH, 0000-0002-2992-2004;
LD, 0000-0002-7076-1871; RNT, 0000-0001-8545-5212

As a countermeasure to the SARS-CoV-2 pandemic, there has been swift development and clinical trial assessment of candidate vaccines, with subsequent deployment as part of mass vaccination campaigns. However, the SARS-CoV-2 virus has demonstrated the ability to mutate and develop variants, which can modify epidemiological properties and potentially also the effectiveness of vaccines. The widespread deployment of highly effective vaccines may rapidly exert selection pressure on the SARS-CoV-2 virus directed towards mutations that escape the vaccine-induced immune response. This is particularly concerning while infection is widespread. By developing and analysing a mathematical model of two population groupings with differing vulnerability and contact rates, we explore the impact of the deployment of vaccines among the population on the reproduction ratio, cases, disease abundance and vaccine escape pressure. The results from this model illustrate two insights: (i) vaccination aimed at reducing prevalence could be more effective at reducing disease than directly vaccinating the vulnerable; (ii) the highest risk for vaccine escape can occur at intermediate levels of vaccination. This work demonstrates a key principle: the careful targeting of vaccines towards particular population groups could reduce disease as much as possible while limiting the risk of vaccine escape.

# 1. Introduction

SARS-CoV-2 has caused a global pandemic with over 115 000 000 reported cases and 2 500 000 confirmed deaths as of 7 March 2021 [1]. In response, multiple vaccine candidates have been rapidly developed, tested in international trials and rolled out in mass vaccination campaigns in many parts of the world [2].

In the UK, two vaccines are in use (as of March 2021), an mRNA-based vaccine produced by Pfizer, and a viral-vectored vaccine produced by AstraZeneca. Phase III trials have determined these vaccines to be highly effective against disease, with the mRNA-based vaccines, in particular, reporting central efficacies against disease (i.e. preventing COVID-19 symptoms) in the range of 94% to 95% [3,4].

With SARS-CoV-2, there remains considerable virological, epidemiological and immunological uncertainty, with implications for vaccine escape currently underdeveloped. In the absence of vaccination, the SARS-CoV-2 virus has demonstrated the ability to mutate and develop variants [5]. Variants with multiple genetic changes have led to phenotypic changes increasing transmissibility [6,7] and mortality [8], and have the potential to reduce the effectiveness of vaccines [5]. The mass deployment of highly effective vaccines, while infection is widespread, may rapidly exert selection pressure on the SARS-CoV-2 virus directed towards mutations that escape the vaccine-induced immune response. However, the strength of this selection and the likelihood of vaccine escape is unknown at this time [9].

Due to limited vaccine supply, countries must decide on priority orders for vaccination. The optimal order of prioritization will depend upon the measure being optimized (i.e. protecting essential societal functions or directly minimizing health harms, such as cases, hospitalizations or deaths, or some combination of these) [10–12]. In the UK, vaccination policy advice is provided by the Joint Committee on Vaccination and Immunization (JCVI). The JCVI advised that the first priorities for the SARS-CoV-2 vaccination programme should be the prevention of COVID-19 mortality and the protection of health- and social-care staff and systems [13]. At the time of the initial prioritization, extremely limited data were available from clinical trials on vaccine efficacy for preventing infection and onward transmission. For the second phase of the vaccination programme, JCVI was asked by the Department for Health and Social Care (DHSC) to formulate advice on the optimal strategy to further reduce mortality, morbidity and hospitalizations from COVID-19 disease. The subsequent advice given was to proceed with an age-based priority order, with operational considerations as part of the justification on account of speed of vaccine uptake being paramount [14].

For prospective investigations, in the absence of empirical data, mathematical models provide a method to gather insight on these questions. We explore the interactions between the deployment of vaccine among the population, infection and disease prevalence and vaccine escape. In this work, we ask the question of how considerations of vaccine escape risk might modulate optimal vaccine priority order. In particular, if infection in vaccinated individuals contributes to pressure to generate vaccine escape, how do the risks depend on the parts of the population that have been vaccinated? Rather than aiming to develop a detailed model of SARS-CoV-2 transmission dynamics, we present a two-population model with differing vulnerability and contact rates to elucidate broad principles on the relationships between epidemiological regimes, vaccine efficacy and vaccine escape. We explore strategies without the constraint of matching the vaccination rollout that has already happened in any country, both for applicability to future scenarios and to other countries.

# 2. Methods

## 2.1. Population heterogeneity

We take the approach of directly building the next generation matrix, based on assumptions about the population and effects of vaccination. We capture population variability in vulnerability and mixing by dividing our model population into two groups: part of the population is more vulnerable to disease and mix less with others, the rest are less vulnerable but mix more with others—as shorthand we term these two groups of the population as 'vulnerable' and 'mixers'. In the main analysis presented here, we make the simplifying assumption that these groups are of equal size: they can be thought of as loosely corresponding to the older and younger halves of the population. Equal proportions are sufficient to illustrate general principles, but the effects of relaxing this assumption are explored in the

Supplementary information (figures S8 and S9), and see also below about how to generalize our approach to any number of groups of any size.

Vulnerability is modelled simply as a ratio $d > 1$ of a higher chance of a severe outcome if a vulnerable individual is infected compared with if a mixer is infected. This might represent progression to hospitalization, need for more intensive treatment or a higher mortality rate. In practice of course, all of these could be separate effects, and 'vulnerability' is not straightforward. However to gain broad insights here, vulnerability is treated in this simple way—a higher chance of a poor outcome if infected, termed 'disease' in the results below for brevity. The more mixing (less vulnerable) half of the population are deemed to have an $m$ times higher rate of contact with others than the rest of the population (all the rest being vulnerable in this model). In terms of a mixing matrix, mixers have $m^2$ higher mixing within their own group than non-mixers have within theirs, and $m$ times higher between groups. To isolate and examine the key factors here of host vulnerability and mixing, we assume that the vulnerable and mixers are equally susceptible to infection, and also equally infectious if infected (only modified by their contact patterns). We also make the assumption in our analysis that there is no prior immunity in this system.

## 2.2. Effects of vaccination

For vaccination, we ignore any delay of effect of vaccination and multiple doses, but we do split the effect of the vaccination into three components. In this model, vaccination can (i) reduce the risk of infection, (ii) reduce the risk of severe disease, and (iii) reduce the risk of infecting others, and we capture these as $\theta_S$, $\theta_D$ and $\theta_I$. These $\theta$ are all separate multiplicative effects on their corresponding rates, and hence $\theta_* = 0$ corresponds to the vaccine having complete/perfect prevention of infection, fully preventing disease given infection or being fully infectivity-blocking and $\theta_* = 1$ means having no effect of the corresponding type. The $\theta_*$ here are comparable to $1 - VE_*$ of Halloran et al. [15].

Translating this framework to a general idea of disease blocking, $\theta_S \times \theta_D$ gives the relative risk of disease for someone vaccinated compared with unvaccinated (so vaccine efficacy in terms of disease blocking would be $1 - \theta_S\theta_D$, while vaccine efficacy in terms of case prevention would be $1 - \theta_S$). For transmission-blocking, it is the combination of susceptibility and infectiousness that matters: $\theta_S \times \theta_I$ gives the relative contribution of population transmission from someone vaccinated compared with unvaccinated. It might be tempting mathematically to use these parameter combinations to reduce this system to two parameters for vaccination, but all three distinct processes are needed to explore the number of vaccinated who become infected, as we argue we should when considering vaccine escape.

## 2.3. Direct calculation

The next generation matrix (NGM) relates the number of infected individuals of each type between infection generations. If the NGM has elements $k_{ij}$, then these entries represent the expected number of new infections in individuals of type $i$ caused by one infected individual of type $j$. Hence, the NGM gives the mapping of distribution of infections between generations [16,17]. Without vaccination, for our model the NGM is proportional to the matrix $\mathbf{M}_0$, given by

$$\mathbf{M}_0 = \begin{bmatrix} 1 & m \\ m & m^2 \end{bmatrix} = \begin{bmatrix} 1 \\ m \end{bmatrix} \begin{bmatrix} 1 \\ m \end{bmatrix}^T, \tag{2.1}$$

where the first population type represents the vulnerable and the second the mixers. Suppose now that a proportion $v_1$ and $v_2$ of the vulnerable and the mixers have been vaccinated respectively. This population can now be thought of as split into four types: the two unvaccinated groups as before (unvaccinated vulnerable, unvaccinated mixers) and then the two corresponding vaccinated groups (vaccinated vulnerable, vaccinated mixers).

Now the NGM is four by four. To understand its construction, first put aside vaccine effects ($\theta_I$ and $\theta_S$). Duplicate the original two by two matrix $\mathbf{M}_0$ horizontally (as without vaccine effects, single unvaccinated or vaccinated individual would cause the same number of secondary infections). Next, split vertically according to vaccine proportions $v_1$ and $v_2$, so row 1 is split into $(1 - v_1)$ times the original row in row 1 and $v_1$ times the original row in row 3 (as without effects of vaccination, those infections caused would be split according to how those populations are split). Including vaccine effects, the third and fourth columns are scaled by $\theta_I$; these groups are proportionately less

infectious. The third and fourth rows are scaled by $\theta_S$; these groups are proportionately less susceptible. Hence, the NGM is proportional to $\mathbf{M}[v_1, v_2]$

$$\mathbf{M}[v_1, v_2] = \begin{bmatrix} (1-v_1) & (1-v_1)m & (1-v_1)\theta_I & (1-v_1)m\theta_I \\ (1-v_2)m & (1-v_2)m^2 & (1-v_2)m\theta_I & (1-v_2)m^2\theta_I \\ \theta_S v_1 & \theta_S v_1 m & \theta_S v_1 \theta_I & \theta_S v_1 m\theta_I \\ \theta_S v_2 m & \theta_S v_2 m^2 & \theta_S v_2 m\theta_I & \theta_S v_2 m^2\theta_I \end{bmatrix}. \tag{2.2}$$

As the original NGM could be written as an outer product, the vaccinated NGM can also be written in this way

$$\mathbf{M}[v_1, v_2] = \begin{bmatrix} (1-v_1) \\ (1-v_2)m \\ \theta_S v_1 \\ \theta_S v_2 m \end{bmatrix} \begin{bmatrix} 1 \\ m \\ \theta_I \\ m\theta_I \end{bmatrix}^T. \tag{2.3}$$

As $\mathbf{M}$ is in the form of an outer product, it has rank one and the spectral radius follows immediately (as the inner product of the same vectors, giving a positive real eigenvalue). The corresponding eigenvector is the column vector of that product. Reading off from equation (2.3), the dominant eigenvector of $\mathbf{M}[v_1, v_2]$ (normalized to have sum of elements equal to one) is

$$\mathbf{P}[v_1, v_2] = \frac{1}{(1-v_1) + (1-v_2)m + \theta_S v_1 + \theta_S v_2 m} \begin{bmatrix} (1-v_1) \\ (1-v_2)m \\ \theta_S v_1 \\ \theta_S v_2 m \end{bmatrix}. \tag{2.4}$$

This vector $\mathbf{P}$ gives the relative proportions of cases as split among the four groups (unvaccinated vulnerable, unvaccinated mixers, vaccinated vulnerable, vaccinated mixers in that order). Furthermore, under general feasible initial conditions (non-negative infections in all groups, not all zero), as all the other eigenvalues are zero the vector denoting the proportion of cases in each group will pivot quickly from any general initial distribution to the dominant eigenvector.

The dominant eigenvalue of $\mathbf{M}[v_1, v_2]$ is

$$\sigma[v_1, v_2] = \begin{bmatrix} (1-v_1) \\ (1-v_2)m \\ \theta_S v_1 \\ \theta_S v_2 m \end{bmatrix} \cdot \begin{bmatrix} 1 \\ m \\ \theta_I \\ m\theta_I \end{bmatrix} = (1-v_1) + (1-v_2)m^2 + \theta_S\theta_I(v_1 + v_2 m^2). \tag{2.5}$$

The effective reproduction ratio is proportional to this dominant eigenvalue $\sigma$ ([17]). To resolve the proportionality, let $R_0$ be the (basic) reproduction ratio when $v_1 = v_2 = 0$. Now we can write the effective reproduction ratio under vaccination, $R[v_1, v_2]$, as

$$R[v_1, v_2] = R[0, 0]\frac{\sigma[v_1, v_2]}{\sigma[0, 0]} = R_0\frac{(1-v_1) + (1-v_2)m^2 + \theta_S\theta_I(v_1 + v_2 m^2)}{1 + m^2}, \tag{2.6}$$

and it is immediately apparent that this equation is linear in the proportions vaccinated ($v_1$ and $v_2$). We note the natural emergence of the transmission-blocking combination of vaccine parameters ($\theta_S\theta_I$) here.

We approximate the effective reproduction ratio as being constant during the period of time under consideration for assessing vaccine effects ($t_{max}$): in other words, there is no susceptible depletion as the timescale is relatively short in terms of the incidence under consideration (the lower the incidence, the longer this period can be). Then, the incidence $y(t)$ is exponential, with growth rate $\lambda$. Again for simplicity, we take $\lambda = (\log R)/T$—the growth rate mapping from $R$ corresponding to a fixed generation time $T$ with no variance. Then the incidence can be easily integrated over time to give the total number of cases during the period in question, and is further simplified by expressing the duration of the period of interest in terms of mean generation time $T$, so $t_{max} = GT$, where $G$ is the duration of the period in terms of disease generations. We will consider the *relative* number of cases below, meaning constants unaffected by changing vaccination can be scaled out. We choose here to scale out initial incidence $y_0$ and also scale by $t_{max}$ (to give $F(R)$ as something that could be interpreted as a time average of cases relative to initial incidence)

$$F(R) = \frac{\int_0^{t_{max}} y(t)\,dt}{y_0 t_{max}} = \frac{\int_0^{t_{max}} y_0\,e^{\lambda t}\,dt}{y_0 t_{max}} = \frac{e^{\lambda t_{max}} - 1}{\lambda t_{max}} = \frac{R^G - 1}{\log(R^G)}, \tag{2.7}$$

for $R \neq 1$. Also, $F(1) = 1$ (either by L'Hôpital's Rule or the integral using $\lambda = 0$). From above, we then have the relative number of cases $C[v_1, v_2]$, compared with a scenario with no vaccination

$$C[v_1, v_2] = \frac{F(R[v_1, v_2])}{F(R_0)},$$

(2.8)

and these cases are distributed among the four subpopulations in proportion to the dominant eigenvector $\mathbf{P}[v_1, v_2]$ given above.

## 2.4. Output metrics

We consider four main outputs. Two are already established above: the effective reproduction rate ($R[v_1, v_2]$) and the relative number of cases ($C[v_1, v_2]$). We define a further two in this section: a measure of the amount of disease relative to no vaccination ($D[v_1, v_2]$) and a measure of vaccine escape pressure ($V[v_1, v_2]$).

For 'disease', we consider the severe outcomes as represented by the vulnerability parameter $d$ (which could represent hospitalization, mortality, or any proxy of interest for severity). We already have the relative number of cases ($C$, equation (2.8)) and know how these are distributed among the four population groups ($\mathbf{P}$, equation (2.4)). The relative risk of disease is multiplied by a factor of $d$ for the vulnerable and $\theta_D$ for the vaccinated (and multiplied by both for the vaccinated vulnerable). For the four respective groups, ordered as previously, the relative risk of disease is proportional to $\mathbf{U}$

$$\mathbf{U} = \begin{bmatrix} d \\ 1 \\ d\theta_D \\ \theta_D \end{bmatrix}.$$

(2.9)

Combining these, we have $D[v_1, v_2]$: a measure of total disease relative to a scenario with no vaccination

$$D[v_1, v_2] = C[v_1, v_2] \frac{\mathbf{P}[v_1, v_2].\mathbf{U}}{\mathbf{P}[0, 0].\mathbf{U}}$$

(2.10)

$$= C[v_1, v_2] \left( \frac{d(1 - v_1) + (1 - v_2)m + d\theta_D\theta_S v_1 + \theta_D\theta_S v_2 m}{(1 - v_1) + (1 - v_2)m + \theta_S v_1 + \theta_S v_2 m} \right) \Big/ \left( \frac{d + m}{1 + m} \right).$$

(2.11)

For 'vaccine escape', reality is a highly complex picture of variants being generated and selected at various scales within and between host [18,19]. Here we take an extremely simple approach and measure pressure on vaccine escape as proportional to the number of cases in vaccinated individuals, treating the vulnerable and mixers as equal in this respect. We also consider sensitivity to including cases in unvaccinated hosts as contributing towards the vaccine escape pressure, see the Supplementary information, figure S2. It is far from clear that our simple approach is the best way to address general questions of vaccine escape, but we propose it here as a straightforward and achievable method. We acknowledge the shortcomings of our approach must be held in mind when interpreting the results.

Building this mathematically, vaccine escape pressure $V[v_1, v_2]$ is modelled as proportional to the number of cases in vaccinated individuals. The normalization for this cannot be the same quantity under no vaccination (this would be a zero denominator), so we use *total* number of cases under no vaccination as the normalization. Let $P[v_1, v_2]$ be the proportion of cases that are in vaccinated individuals

$$P[v_1, v_2] = \mathbf{P}[v_1, v_2].\begin{bmatrix} 0 \\ 0 \\ 1 \\ 1 \end{bmatrix} = \frac{\theta_S v_1 + \theta_S v_2 m}{(1 - v_1) + (1 - v_2)m + \theta_S v_1 + \theta_S v_2 m}.$$

(2.12)

Then $V[v_1, v_2]$ is the product of the relative cases ($C[v_1, v_2]$) and the proportion of these cases that are in vaccinated individuals ($P[v_1, v_2]$)

$$V[v_1, v_2] = C[v_1, v_2] P[v_1, v_2]$$

(2.13)

$$= C[v_1, v_2] \frac{\theta_S v_1 + \theta_S v_2 m}{(1 - v_1) + (1 - v_2)m + \theta_S v_1 + \theta_S v_2 m}.$$

(2.14)

## 2.5. Extension to more general population structures

It is straightforward to generalize to $n$ population groups, where group $i$ has relative size $x_i$ of the population, a relative vulnerability $d_i$ and relative mixing $m_i$ (with one degree of freedom in each of these, so either the value for one group can be set to unity, or the total can be normalized). When considering more general population structures, relative susceptibility to disease or infectiousness to others can also be included ($\mu_i$ and $\tau_i$, respectively)—this may be particularly important if the population is broken down into age classes considering children separately.

Following analogously from above, the next generation matrix is of size $2n \times 2n$ and can be written as an outer product

$$
\text{NGM} = \begin{bmatrix} (1-v_1)x_1\mu_1 m_1 \\ (1-v_2)x_2\mu_2 m_2 \\ \vdots \\ (1-v_n)x_n\mu_n m_n \\ \theta_S v_1 x_1 \mu_1 m_1 \\ \theta_S v_2 x_2 \mu_2 m_2 \\ \vdots \\ \theta_S v_n x_n \mu_n m_n \end{bmatrix} \begin{bmatrix} m_1\tau_1 \\ m_2\tau_2 \\ \vdots \\ m_n\tau_n \\ \theta_I m_1 \tau_1 \\ \theta_I m_2 \tau_2 \\ \vdots \\ \theta_I m_n \tau_n \end{bmatrix}^T .
\tag{2.15}
$$

As before, this is a rank one matrix and the spectral radius here is the inner product of the same vectors, giving the proportionality with the effective reproduction ratio $R$

$$
R[\mathbf{v}] = R_0 \frac{\sum_{i=1}^{n} ((1-v_i) + \theta_I \theta_S v_i)x_i\mu_i m_i^2 \tau_i}{\sum_{i=1}^{n} x_i\mu_i m_i^2 \tau_i},
\tag{2.16}
$$

where $\mathbf{v} = (v_1, v_2, \ldots, v_n)$. The calculation for the relative number of cases $C$ in terms of $R$, relative to a scenario with no vaccination, is exactly as in equation (2.8). The distribution of cases, $\mathbf{P}$, also follows as being the dominant eigenvector normalized to sum to 1

$$
\mathbf{P}[\mathbf{v}] = \frac{1}{\sum_{i=1}^{n} ((1-v_i) + \theta_S v_i)x_i\mu_i m_i} \begin{bmatrix} (1-v_1)x_1\mu_1 m_1 \\ (1-v_2)x_2\mu_2 m_2 \\ \vdots \\ (1-v_n)x_n\mu_n m_n \\ \theta_S v_1 x_1 \mu_1 m_1 \\ \theta_S v_2 x_2 \mu_2 m_2 \\ \vdots \\ \theta_S v_n x_n \mu_n m_n \end{bmatrix} .
\tag{2.17}
$$

The measure of total disease relative to a scenario with no vaccination follows similarly

$$
D[\mathbf{v}] = C[\mathbf{v}] \frac{\sum_{i=1}^{n} d_i((1-v_i) + \theta_D \theta_S v_i)x_i\mu_i m_i}{\sum_{i=1}^{n} ((1-v_i) + \theta_S v_i)x_i\mu_i m_i} \Bigg/ \left( \frac{\sum_{i=1}^{n} d_i x_i\mu_i m_i}{\sum_{i=1}^{n} x_i\mu_i m_i} \right).
\tag{2.18}
$$

The vaccine escape pressure is again a product of the relative cases and the proportion of these cases that are in vaccinated individuals

$$
V[\mathbf{v}] = C[\mathbf{v}] \frac{\sum_{i=1}^{n} \theta_S v_i x_i\mu_i m_i}{\sum_{i=1}^{n} ((1-v_i) + \theta_S v_i)x_i\mu_i m_i} .
\tag{2.19}
$$

Further generalizations are possible within this analytic framework, for example the vaccine effects $\theta_S$, $\theta_I$ and $\theta_D$ could vary by age group, though this would require additional parametrization. In the more general case that the mixing structure cannot be written as an outer product then it is likely a numerical approach would be needed.

## 2.6. Parametrization

The goal of our simple modelling approach is to gain general results which hold true both in more complex models and in broad realistic ranges of parameters, and therefore give valuable insights. Hence, a detailed parametrization here is not required, but we can base our parameters in 'ballpark' ranges corresponding to current knowledge.

For vaccination parameters, knowledge is currently growing at a pace on vaccine effectiveness. From clinical trials of the Pfizer vaccine, using data for those cases observed between day 15 and 28 after the first dose, efficacy against symptomatic COVID-19 has been independently estimated by Public Health England as 91% (74 to 97%) [20]. Assessment of clinical trial data for the Oxford/AstraZeneca vaccine has shown vaccination (two standard doses given 12 or more weeks apart) to reduce symptomatic disease by 81.3 (60.3–91.2%); while protection following the first dose is estimated as 76.0% (59.3–85.9%) between days 31 and 60. The level of protection against infection (both symptomatic and asymptomatic) was found to be 63.9% (46.0–76.9%) after one dose and 59.9% (35.8–75.0%) after two doses [21].

We are beginning to see real-world evidence of vaccine effectiveness through observational studies. Against symptomatic COVID-19 in older people in the UK, one observational study found that a single dose of the Pfizer vaccine was approximately 60–70% effective at preventing symptomatic disease in adults aged 70 years and older in England, and two doses were approximately 85–90% effective. The effect of a single dose of the Oxford/AstraZeneca vaccine against symptomatic disease was approximately 60–75% [22]. Estimates of the likelihood of severe outcomes conditional on symptomatic infection have also been gathered. For the Pfizer vaccine, those aged 80+ and vaccinated who went on to become a symptomatic case had a 43% lower risk of hospitalization (within 14 days of a positive test) and a 51% lower risk of death (within 21 days of a positive test) compared with unvaccinated cases. The effect of a single dose of the Oxford/AstraZeneca vaccine in those aged 80 and above who went on to become a symptomatic case was 37% protection against hospitalization within 14 days of a positive test [22]. More recent results show protection against hospitalization from a single dose of either the Oxford/AstraZeneca or Pfizer vaccines to be around 80% [23].

The picture of the capability of the available vaccines to prevent onward transmission is currently less clear. Ascertaining the magnitude of any transmission-blocking effect most directly will require detailed observational studies in closed settings or households. All of these could be further complicated by age-dependencies, such as the rate of hospitalization [24], and further disparities in case and severe outcomes due to pre-existing health conditions and socio-demographic factors [25]. As well as refinement of estimates over the coming months, vaccine effects may be modulated in the face of new variants in future.

For our default vaccination parameters we take $\theta_S = 0.6$, $\theta_I = 0.6$, $\theta_D = 0.3$. This corresponds to a relative risk of disease of $\theta_S \times \theta_D = 0.18$, comparable with a vaccine effectiveness of around 80%. Transmission-blocking is perhaps the most uncertain factor here, and our values correspond to $\theta_S \times \theta_I = 0.36$—transmission reduced by a factor of around 3. Transmission assumptions are key to the resulting dynamics, and our knowledge of appropriate parameters here may change in the near future, so sensitivity to this is explored below (figure 2) and further in Supplementary information, figures S1, S4, S6.

For the population heterogeneity, the two groups of vulnerable and mixers could be thought of as loosely corresponding to older and younger age groups, though here we are not considering children whose mixing patterns and also their susceptibility and infectiousness for SARS-CoV-2 could be very different to those of adults [26–28]. To approximate a 'mixing' parameter, the BBC pandemic study [29], with data from the UK in 2017–2018, shows the mean number of contacts by age. While there are clear differences by age, the ratios are not large. A visual inspection of younger adults versus older adults, allows us to approximate the range for $m$ as 1–2 by default.

The vulnerability ratio $d$ is not straightforward to parametrize as: (i) we are using this to explore severe outcomes in an abstract way, so it could correspond to probability of hospitalization or a case fatality ratio or any other measure of severe disease, and (ii) the simple two-population structure is for exploration of the effects of heterogeneity rather than explicitly corresponding to defined population groups. Furthermore, estimates for COVID-19 severity vary between studies, depending on context [30–32] and the presence of more pathogenic variants [6,8].

Note that many of our results are not dependent upon the vulnerability ratio $d$ ($R$, $C$ and $V$ are all independent of $d$), but relative disease $D$ and thus consideration of strategies to best avert disease of course will depend on $d$. Below, we have taken $d = 10$ as the default in plots to explore the case where the vulnerable group is substantially more at risk, but the other half of the populations cannot be neglected for disease risk. For results on disease below, these are shown for a range of values of $d$ (1 to 10). Our results are not sensitive to taking even higher values of $d$—we explore this in the Supplementary information.

For the parameters for the scenario under consideration, we have considered a situation where $R > 1$ initially before vaccination, choosing particularly $R_0 = 1.2$ which approximately corresponds to mid-September and October 2020 in UK [33]. At that time, some regions were under tight restrictions and

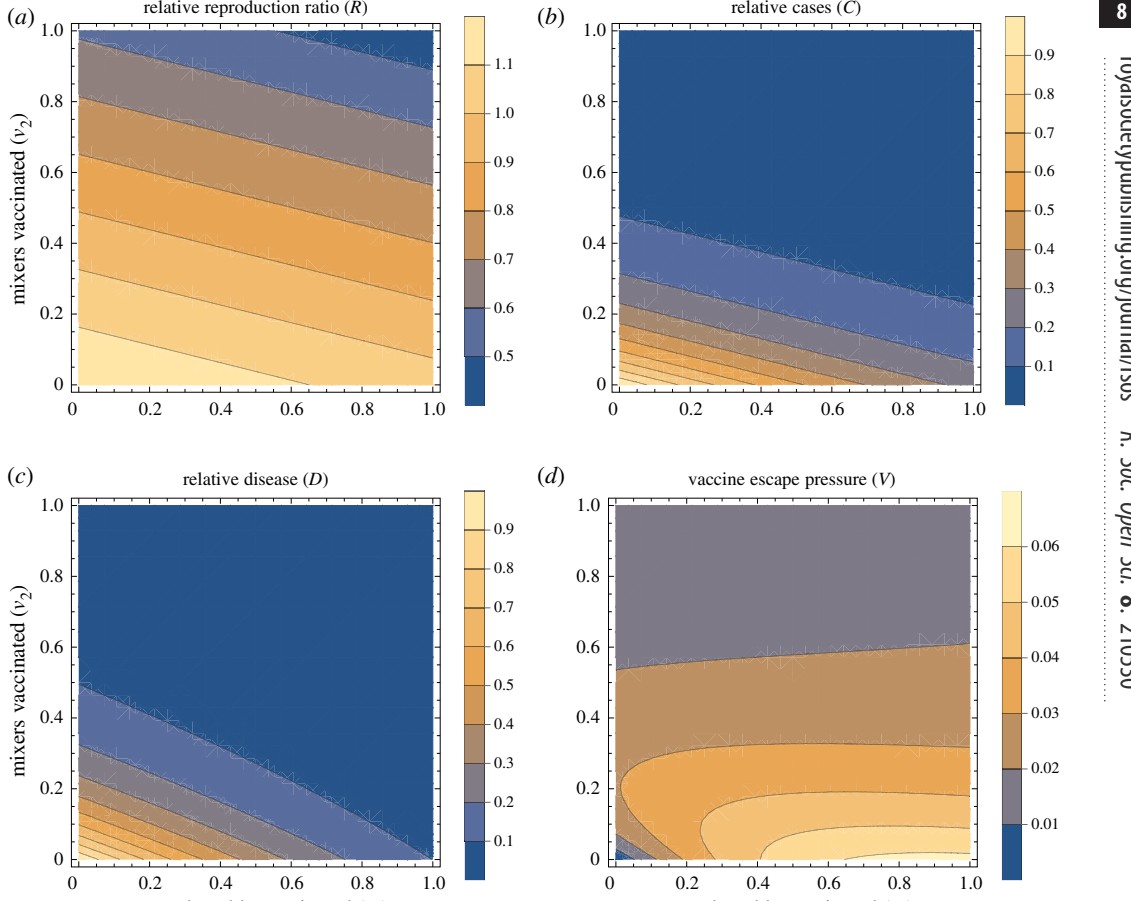

**Figure 1.** Summary outputs. Summary outputs for a fixed set of population parameters ($m = 2$, $d = 10$), vaccine parameters ($\theta_S = 0.6$, $\theta_I = 0.6$, $\theta_D = 0.3$) and scenario under consideration ($R_0 = 1.2$, $G = 15$). Four output measures are shown, relative reproduction ratio $R[v_1, v_2]$ ($a$), relative cases $C[v_1, v_2]$ ($b$), relative disease $D[v_1, v_2]$ ($c$), and vaccine escape pressure $V[v_1, v_2]$ ($d$). All four panels are shown as contour plots with horizontal and vertical axes representing the proportion of vulnerable and mixers vaccinated ($v_1$ and $v_2$, respectively).

some interventions were in place everywhere. We note that the choice $R_0 = 1.2$ is clearly not a true $R_0$, but instead here $R_0$ is termed for the value of the effective reproduction ratio at this time if there was no vaccination. The value of $G$, the time period considered as measured in mean generation times, is going to be a subjective decision. Estimates for the generation time are variable between studies, but typically around 4–6 days [34–36]. We take $G = 15$ by default, corresponding to a time window of two to three months. How results vary with $G$ is discussed below, and $G = 5$ is used as an example to show how outputs change with a shorter $G$ in Supplementary information, figures S3–S6.

## 3. Results

### 3.1. Dependency of epidemiological outcomes on vaccine coverage

We show summary results for typical parameters in figure 1. The effective reproduction ratio decreases as more people are vaccinated (figure 1$a$). From the analytic expression above, we can see that this decrease in the effective reproduction ratio occurs for all parameter values so long as there is any transmission-blocking effect of the vaccine ($\theta_S \theta_I < 1$). Furthermore, the dependence on the proportion vaccinated is linear, with stronger effect (by factor $m^2$ here) for vaccinating the mixers. The cases (figure 1$b$) are here a direct function of $R$ so also decrease with increasing vaccination, but not linearly: there is a steep drop to $R = 1$ and thereafter the effect is smaller, simply reflecting prevalence dropping faster during the period in question. Intuitively, we expect similar vaccine effects on $R$ and total cases will hold in more complex models.

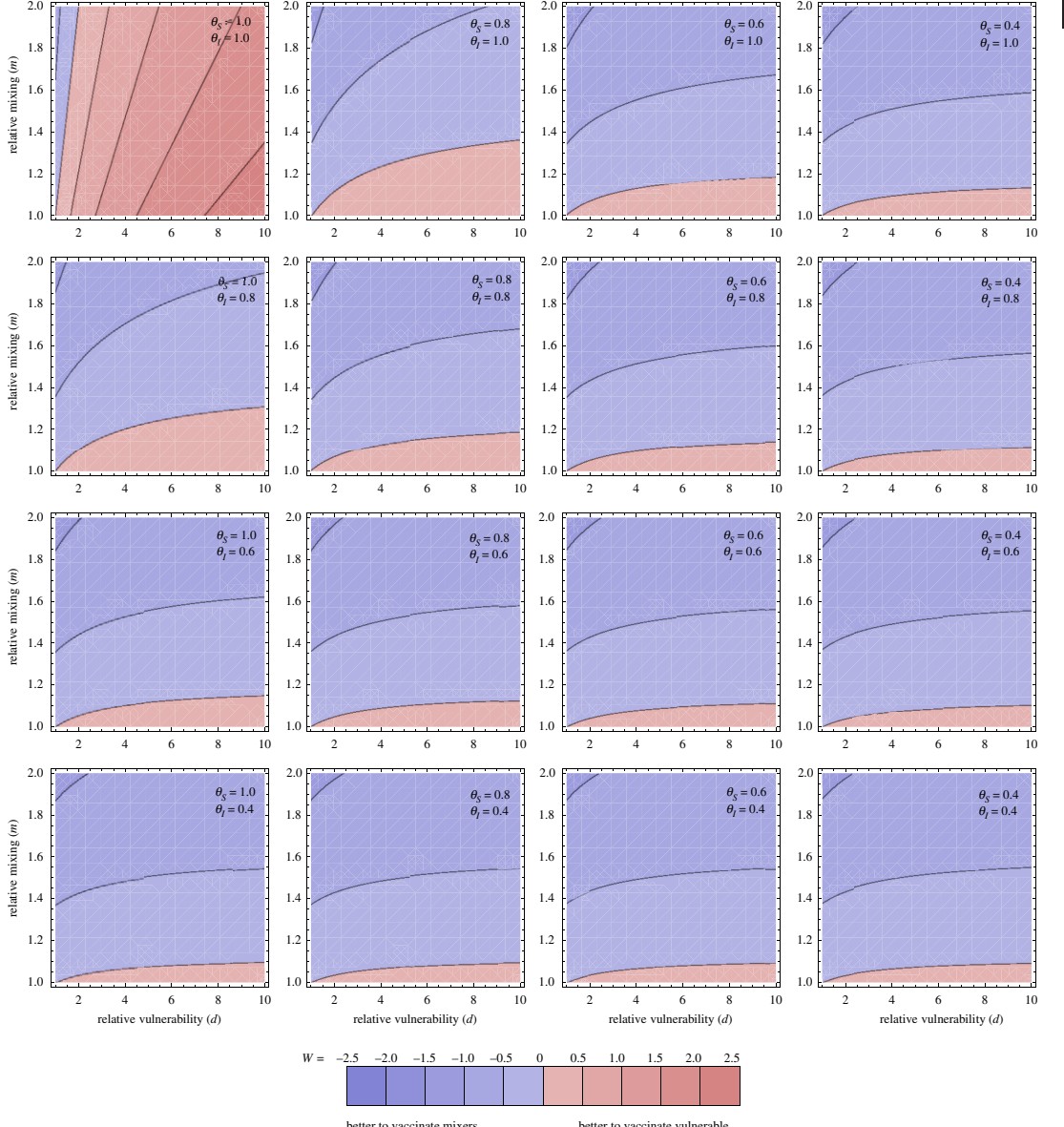

**Figure 2.** Optimal allocation for averting disease. Optimal allocation for averting disease (as measured by $W$: logged ratio of disease averted by vaccinating the vulnerable compared with the mixers). Individual panels explore the population parameters (vulnerability $d$ and mixing $m$ on horizontal and vertical axes). The contour values are kept fixed between the plots, with the contour for ratio 1 between the blue and red, and meeting the bottom left of every panel (where $m = d = 1$ so the population is homogeneous). More disease is averted by vaccinating the vulnerable than the mixers in the pink regions and vice versa in the blue regions. Different panels vary the effects of the vaccine: the rows step through $\theta_I = 1$, 0.8, 0.6, 0.4 and the columns step through $\theta_S = 1$, 0.8, 0.6, 0.4. Through all panels, $\theta_D = 0.3$. Thus the top left panel corresponds to the vaccine having no transmission-blocking effects at all, and stepping right and down increases transmission-blocking through reduced susceptibility or infectivity. $\epsilon = 0.1$. All other parameters are as in figure 1.

## 3.2. Effect of vaccine on number of cases with severe disease

The total number of severe infections over this fixed period, denoted here as disease (figure 1c), decreases as vaccination increases in either group. However, this is no longer purely a function of $R$: it is also dependent on *who* is infected—the distribution of cases among the vulnerable and mixers. If vaccination coverage is higher in the vulnerable than the mixers, disease is disproportionately brought down relative to cases, and this is visible as a slight curve of the contours in the bottom right of the panel (where $v_1$ is high and $v_2$ is low).

It is intuitive that for a very wide range of models, vaccinating more people in any group has the effect of decreasing cases in that group and also possibly other groups, driven by the dual effects of vaccination in

transmission-blocking and disease-blocking. The question remains of which group it would be most effective to vaccinate to reduce severe disease (or any other outcome represented by increased vulnerability).

This question of optimal allocation can be explored by considering a situation in which we have a fixed amount of vaccine (enough to vaccinate a proportion $\epsilon$ of either group, say), and considering either solely vaccinating the most vulnerable or the mixers and evaluate $W$, the logged ratio of disease prevented by vaccinating the vulnerable compared with vaccinating the mixers

$$W = \log\left(\frac{1 - D[\epsilon, 0]}{1 - D[0, \epsilon]}\right), \tag{3.1}$$

where $D[v_1, v_2]$ is the relative total disease function set up above (where $D[0, 0] = 1$).

In figure 2, $W$ is explored as a function of vulnerability of the vulnerable ($d$), mixing of the mixers ($m$) and the two transmission-blocking effects of the vaccine ($\theta_S$ and $\theta_I$). Here we set the proportion $\epsilon = 0.1$, but given that disease is near linear in $v_1, v_2$ it will not be very sensitive to this. The overall picture is that in the majority of the parameter space explored, vaccinating the mixers is more effective than vaccinating the vulnerable to reduce the total amount of disease.

This might not be intuitive—intuition may say to focus vaccination on the vulnerable. The result here hinges on the transmission-blocking effects of the vaccination dominating: bringing down $R$ overall means fewer cases in the vulnerable and the most efficient way to do that is to vaccinate the mixers. There are three edges of parameter space, each discussed below, where this effect is reversed: (i) where there is little difference in mixing between the groups ($m$ is close to one), (ii) when there is no (or very little) transmission-blocking effect ($\theta_S = \theta_I = 1$), or (iii) when the time horizon that we are optimizing over is very short ($G$ small).

For (i), $m$ is close to 1, this is visible just above the horizontal axis in the individual panels in figure 2. In this case, as $m \approx 1$, the 'mixing' half of the population is not actually so different from the vulnerable half in terms of their role in population transmission, and the benefits of vaccinating them are reduced. This could happen if there was little heterogeneity in mixing to start with, or the vulnerable started to mix more as the vaccine rolled out. This also can happen analogously when the population proportions are varied so the vulnerable are a small group, and mixing is largely uniform in the rest of the population (figures S8 and S9 in the Supplementary information).

For (ii), if the vaccine is not transmission-blocking but purely disease-blocking, then it makes sense that the only use of the vaccine is the direct benefits of protecting the individuals vaccinated, rather than any impact on the epidemic trajectory. The top left panel of figure 2 shows this effect, but also illustrates the exception within this exception (the blue wedge along the vertical axis). When there is strong mixing in the mixing group, then cases are disproportionately in that group. Even though those in the mixing group are less likely to develop severe disease if they become infected, the much higher rates of infection in mixers mean that the vaccine is still best deployed to directly protect the mixing group. Under this simple two-population model, this will be when $m > d$ (which can be seen from the distribution of cases determined by the eigenvector above).

For (iii), shifting to a shorter time window means that the change to the epidemic trajectory induced by the vaccine becomes less important as the focus is on more immediate effects. This is explored in the Supplementary information. In the extreme, this will become like case (ii) above: the distribution of cases in the groups must be weighted against the relative vulnerability so $d > m$ again for it to make sense to vaccinate the vulnerable preferentially.

Overall, the results in this model show that the effects of vaccination on reduction of cases can give a counterintuitive optimal strategy: vaccinate the mixers to best protect the vulnerable. This result in the present model is chiefly driven by the dynamic trajectory of the epidemic responding to transmission-blocking effects of vaccination, but also slightly by the burden of infection being disproportionately among the most mixing part of the population. The generality or otherwise of this result is discussed below, and this result must be viewed together with the caveats to this simple approach, also discussed below.

## 3.3. Vaccine escape

As described above, we represented vaccine escape pressure in the simplest way as the number of cases in vaccinated individuals. Even for this simple model approach, a rich picture emerges (figure 1d). With none of the mixers vaccinated, vaccinating more vulnerable mostly just increases vaccine escape pressure. However, this is not true the other way around: with no vulnerable vaccinated, then vaccinating the mixers at first increases vaccine escape pressure, and later decreases for greater vaccine uptake among the mixers population. This result can be interpreted intuitively: increasing vaccination of mixers

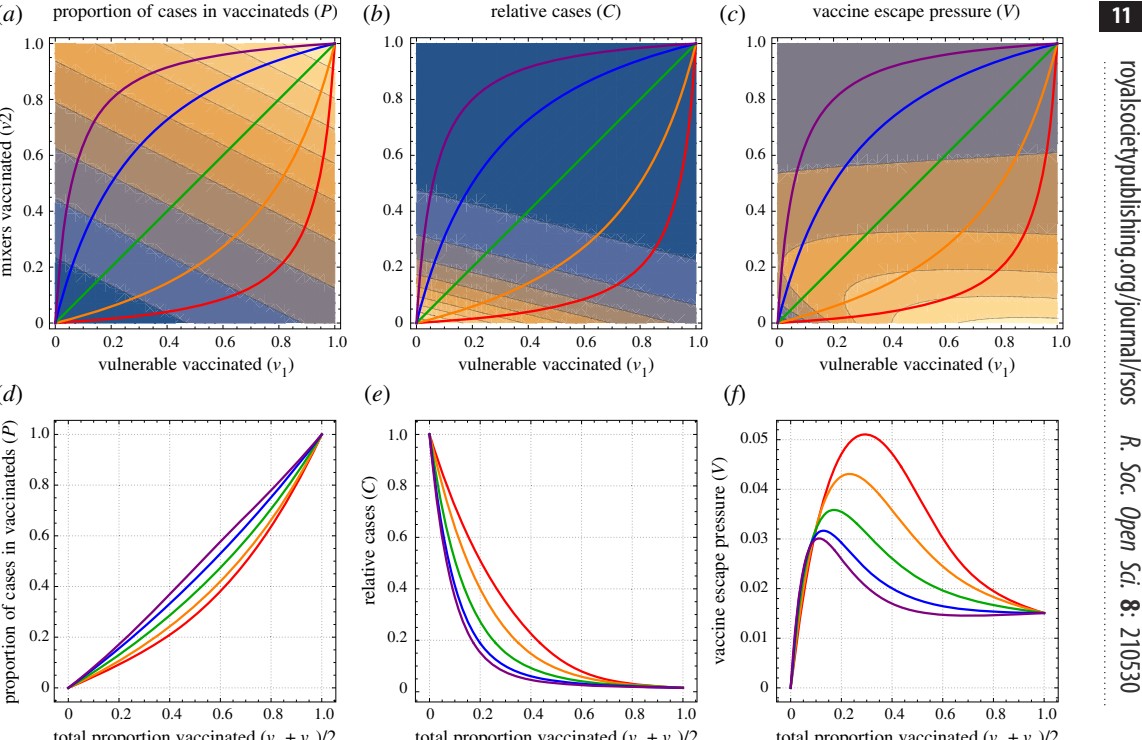

**Figure 3.** One-dimensional paths to explore vaccine escape. The left pair of plots show the proportion of cases that are among those vaccinated, the middle pair give the total number of cases (relative to if there was no vaccination) and the right pair give the vaccine escape pressure. All parameters are as in figure 1. The top row shows all of these as functions of the proportion of vulnerable and mixers vaccinated ($v_1$ and $v_2$, respectively, on horizontal and vertical axes). The coloured lines show five one-dimensional paths, as the total number vaccinated varies from none to all of the population, taking different routes in terms of the mix of vulnerable and mixers. The lower plots correspond to outputs on those one-dimensional paths. The proportion of cases in vaccinated people increases as a function of the proportion vaccinated, while the total number of cases decreases. The product of these gives a measure of vaccine pressure that is maximal for intermediate levels of vaccination.

increases the *proportion* of cases who are vaccinated, but decreases the overall *absolute* number of cases. These two effects combine to give a maximum at intermediate levels of vaccination. This is explored over a wider range of vaccine parameters in Supplementary information, figure S1—the same effect holds except when the vaccine has no transmission-blocking effects.

The non-monotonic effects are investigated further in figure 3 by considering one-dimensional lines from no vaccination to full vaccination, varying in terms of path taken in terms of balance of vulnerable and mixers. In all of these, the total cases $C$ decreases with more vaccination (figure 3$e$), but the proportion of these cases which are in those who have been vaccinated ($P$) increases with more vaccination (figure 3$d$). The product of these ($C \times P$) gives the vaccine escape pressure, which is unimodal: there is highest risk at some intermediate range of vaccination. This peak is highest by vaccinating vulnerable first, but the peak is there for all paths for the parameters used here.

These effects are dependent on the vaccine changing the trajectory of the epidemic and bringing cases down. For a shorter time horizon, there is less time for these effects to come into play. Similarly, if the cases in unvaccinated individuals also contributed vaccine escape, then our results could be modified, in particular to remove the region of low pressure for low vaccination. This is intuitive, and could also be seen as linearly interpolating from vaccine escape pressure as shown in figure 1$d$ towards purely the count of cases in figure 1$b$. Both of these sensitivities are explored further in the Supplementary information.

# 4. Discussion and conclusion

## 4.1. Summary

There are multiple facets to consider when determining a prioritization order for delivery of a limited vaccine supply. Here we suggest that pressure on vaccine escape should be part of these considerations, and that

exploratory modelling can highlight where the risk points are. By analysing a simple model of two populations with differing vulnerability and contact rates we unpick combinations of epidemiological regimes and vaccine efficacy where the risk of vaccine escape is heightened.

Our results illustrate two main insights: (i) vaccination aimed at reducing infection prevalence could be more effective at reducing disease than directly vaccinating the vulnerable, and (ii) the highest risk for vaccine escape can occur at intermediate levels of vaccination. In particular, vaccinating most of the vulnerable and few of the mixers could be the most risky for vaccine escape.

## 4.2. Caveats and areas for further development

By the very nature of the model being a simple representation of a complex system, there are numerous associated caveats to our approach. We restricted our main analysis to only two types of heterogeneity (vulnerability and mixing). In reality, there are many different risk factors affecting transmission dynamics and vaccine uptake, such as age-dependent susceptibility and infectivity. However, we explored two types of heterogeneity alone in order to assess their effects in as simple a setting as possible, without the effects of additional factors. Furthermore, we considered the population split into equal halves. This is relaxed somewhat in further work in the Supplementary information, in which we show that our main results are robust to this assumption. But a more realistic structure will involve more than two population groups—we outlined above how the analytical framework may be extended to more general population structures.

Even extrapolating from the insight that vaccinating mixers first may be optimal for both reducing disease and vaccine escape risk leaves the question of who those mixers are in practice. The group most central to transmission might not simply be a function of age. For example occupation could be taken into account, e.g. those whose roles necessitate contact with others. Another important dimension could be household structure, e.g. those who live with several other people. The interplay between mixing and vulnerability is also important, for example the epidemiological bridging roles played in connecting the most at risk to the wider community by healthcare workers, and household members of the extremely vulnerable.

Another simplification here is that we considered an epidemic in a single population. In reality, the risk of vaccine escape in any population depends not only on the possibility of vaccine escape variants arising locally, but also on the possibility of such variants being imported from elsewhere. Studies that seek to design vaccine strategies based on a range of objectives might also consider the risk of vaccine escape variants being imported when deciding how vaccines should be prioritized. Nonetheless, we contend that minimizing the risk of vaccine escape locally should be a component of any objective function involving vaccine escape.

A further area of substantial oversimplification in the approach presented here is in the mechanism of vaccine escape, and specifically where generation and selection of escape mutants occurs. In practice, mutation, competition and selection will be operating at both within host and between hosts, which poses considerable challenges for capture by models [37]. Here we simply consider when, in terms of vaccination regimes, the pressures (selection within- and between-host combined) may be greatest, by considering transmission to vaccinated hosts. Though this is slightly relaxed to consider unvaccinated hosts also contributing to vaccine escape pressure in the Supplementary information, this approach is clearly still very crude. An approach which included the circulation of any escape variants would need to develop assumptions about the dynamic effect of these variants, e.g. to what extent would variants abrogate the different vaccine effects of susceptibility, infectivity and disease reduction. An extreme approach, where vaccination is perfect against wild-type but completely ineffective against an escape variant, found that establishment of the resistant strain was most likely when most of the population had been vaccinated [38].

A key assumption running through the approach here is that the effects of the vaccine feed through to reshape the overall epidemic, whether this is by design, or an unplanned benefit from a vaccine which is unexpectedly transmission-blocking. An alternative to this would be if non-pharmaceutical interventions (NPIs) are adaptive to prevalence and observed epidemic patterns, for example adjusting to keep the effective reproduction ratio just below 1, or prevalence below some target. In this case, the optimal allocation of the vaccine would no longer be controlling the epidemic directly, but should instead account for the level of NPIs that are needed along the way, where one of the objectives may be to minimize NPIs to mitigate their wider costs and harms. Furthermore, the proportion protected by the vaccine is kept fixed during the period under consideration—a more realistic model of ongoing phased vaccine rollout would be warranted particularly in the context of a more detailed model of population heterogeneity as discussed above.

We made simplifying assumptions on implementation of vaccination to aid analytical tractability. Our approach does not address at all the kinetics of vaccine protection developing in the days/weeks following inoculation. We treated vaccination as a single dose vaccine, with the impact of two doses and dosage spacing a candidate for future research. In reality, we recognize this is a simplified representation of a complex process, whereby new supplies of vaccine are being manufactured and distributed over time, where second dose efficacy may change depending on the inter-dose separation, and that there can be an intrinsic feedback between vaccination rates and population level incidence. We also have not considered any waning in immunity, either that induced by infection or from receiving a vaccine. These and related partial immunity effects are areas which urgently require further attention, particularly in terms of addressing implications for vaccine escape [39–43].

Despite these caveats, the model considered here, which includes important features of transmission and vaccination, enabled us to illustrate the key principle that the careful targeting of vaccines towards particular groups allows case numbers to be reduced while limiting the risk of vaccine escape. We hope that the proposal of general principles under this abstracted system will motivate further investigation under more detailed models.

## 4.3. Relation to classic theory and recent results

Our model demonstrating that intermediate levels of vaccination could be the highest pressure for vaccine escape is resonant with established theory. In Grenfell *et al.*, in a phylodynamic model of an individual host, adaptation was highest at intermediate levels of immunity, driven by a maximal combination of viral abundance and strength of selection [44]. In the context of SARS-CoV-2, these favourable circumstances for antigenic evolution at the host level have been observed during prolonged COVID-19 infections in an immunocompromised individual [45]. Our population-level result here is analogous, with total infections playing the role of viral abundance and proportion of infections in vaccines playing the role of strength of selection. The importance of host heterogeneity in driving this maximal pathogen escape pressure has also been described in a bacteria and bacteriophage system [46].

Our study adds to a growing knowledge base on the potential of emergence of variants with vaccine escape properties under the circumstances of widespread infection prevalence and different dosing regimen. An immuno-epidemiological model found that under certain scenarios a one-dose policy may increase the potential for antigenic evolution; specifically, a vaccine strategy with a very long inter-dose period could lead to marginal short-term benefits (a decrease in the short-term burden) at the cost of a higher infection burden in the long term and substantially more potential for viral evolution [41]. However, it has been argued that so long as vaccination provides some transmission-blocking effects, the corresponding reduction in prevalence should more than counterbalance concerns about antigenic escape pressure from delaying a second dose [18].

Limited vaccine supply has necessitated policymakers requesting advice on the priority order for SARS-CoV-2 vaccines. This guidance has had to be offered in the presence of limited data, with an expectation that additional knowledge would subsequently be accrued on vaccine efficacy for preventing infection. In the UK, dynamic infectious disease transmission models have been a contributor to the decision-making process, with the advised ordering primarily going in descending age order [47,48].

The result here that vaccinating mixers would be more effective to reduce severe disease than vaccinating the vulnerable for the majority of the reasonable parameter range for our model is in contrast to Moore *et al.*, where vaccinating the oldest first was consistently the best approach to minimize deaths and disease [47]. There are a number of assumptions that differ between the two approaches, including vaccine effects and different population heterogeneity patterns. We are also considering here a vaccine rollout during higher prevalence (as opposed to vaccination before a possible next wave) and a different time period is under consideration. It is not clear which combination of these differences are key, but probably it will fundamentally come down to the relative utility of the vaccine in reducing overall prevalence versus directly protecting the most vulnerable. Further work is needed to unpick these differences, and promising directions include exploring the assumed distributions of vulnerability and mixing among the population (see Supplementary information).

Speculatively, it is possible that with more of a spectrum of population heterogeneity the optimal strategy for mitigating both disease and the risk of vaccine escape could involve something like first vaccinating the most extremely vulnerable to immediately protect them, then pivoting to the core mixers to bring down prevalence and later back to vaccination of the moderately vulnerable. It is also likely that the optimal strategy in that scenario will depend on the rate of vaccine availability.

The key advance from our approach over others is that it has brought in considerations of vaccine escape pressure, albeit in crude form, together with also considering overall infection and disease rates in a heterogeneous population. However, our model is relatively simple. While this has allowed us to uncover broad insights, further explorations in more complex models will establish if the qualitative results are robust to including more realistic detail. We recommend that vaccine escape risks should be included as part of considerations for vaccine strategies, and that further work is urgently needed here.

Data accessibility. The data are provided in electronic supplementary material.

Authors' contributions. Conceptualization J.R.G., E.M.H., L.D., R.N.T. Formal analysis J.R.G. Methodology J.R.G., E.M.H., L.D. and R.N.T. Software J.R.G. Visualization J.R.G. Writing original draft J.R.G., E.M.H. Writing review and editing J.R.G., E.M.H., L.D. and R.N.T.

Competing interests. We declare we have no competing interests.

Funding. This work was supported by UKRI through the JUNIPER modelling consortium (grant no. MR/V038613/1)

Acknowledgements. The authors thank Angela McLean for valuable discussions, and Matt Keeling, Deirdre Hollingsworth and two anonymous referees for helpful feedback.

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
