## [Peer Review File · Royal Society Open Science]

Review History

RSOS-210530.R0 (Original submission)

Review form: Reviewer 1

Is the manuscript scientifically sound in its present form?

Yes

Are the interpretations and conclusions justified by the results?

Yes

Is the language acceptable?

Yes

Do you have any ethical concerns with this paper?

No

Have you any concerns about statistical analyses in this paper?

No

Recommendation?

Accept with minor revision (please list in comments)

Comments to the Author(s)

The manuscript by Dr. Gog and colleagues deals with the analysis of a SIR-like epidemiological model applied to the transmission of SARS-CoV-2. Using the model, the authors discuss several vaccination strategies for a population composed of subgroups characterized by different mixing and vulnerability patterns. The focus of the analysis, besides the derivation of standard epidemiological metrics such as the reproduction number and the incidence of infection, is on the possibility for vaccines to exert selection pressure on the virus, ultimately resulting in the emergence of mutations that may be able to escape the immune response triggered by the administration of the vaccine.

Needless to say, the topic is of extreme interest. The almost equation-free approach used by the authors may also serve well the purpose of widening the readership of an otherwise technical manuscript. The toy-like nature of the model seems to be better suited to seek general mechanisms rather than specific decision-making prescriptions. This point is effectively addressed in the manuscript and should not be seen, in my view, as a limitation of this study. The presented results seem sound, given the hypotheses laid out by the authors.

That being said, I have some technical comments that the authors may want to consider while revising their work:

- The complexity of the model analyzed in the main text is kept to a minimum---and, I would argue, understandably so. Of the several simplifying hypotheses that have been introduced, one leaves me a bit perplexed, though: namely, that the two groups have the same relative abundance within the population. Besides the obvious unlikelihood of such numerical coincidence, I wonder whether this choice could perhaps lead to an underestimation (not quite by the authors, rather by some readers) of possible asymmetries in the transmission process and in the definition of epidemiological patterns. I am especially referring to the analytical treatment, where the 'vulnerable'-to-'mixer' ratio is nowhere to be found, exactly because of this strong 1-to-1 hypothesis. However, this ratio influences several of the results presented in this work, as acknowledged (and even shown) by the authors. I would suggest removing this 1-to-1 hypothesis from the main text while keeping all the other simplifications in place. Numerically, I would not change anything, meaning that the main text could still just account for the case $\epsilon=1$ (borrowing from the extended model presented in the main text).

- I am not against the 'direct calculation' approach chosen by the authors for the definition of the next-generation matrix. However, equation (2) needs to be better framed and more explicitly explained to make sure that readers can easily follow. For instance, I believe that at least some future readers might be left somehow dumbfounded by the fact that the fraction of vaccinated infectious people does not appear in the last two entries of the first row of the next-generation matrix (similar remarks apply to other entries as well). A similar observation holds also for equations (3) and (4), which are introduced basically with no prior methodological background. I believe that in all these cases the authors would do the less mathematically-versed readers a solid if they could expand just a bit the explanation of these technical aspects of their work.

- Some epidemiological terms need to be better defined. For instance, I cannot fully understand what do the authors mean when they say that in their model "incidence $I(t)$ is exponential, with growth rate λ " (p.4, l.39). Now, if they have in mind a model like $dI/dt=\lambda I$, then I guess that $I(t)$ would be the cumulative incidence at time t ; if so, I do not get where the integral in equation (5) comes from. Some further explanation seems to be warranted here. The same goes for the term "prevalence", which seems to be used naively (in both the abstract and the summary).

- Selection pressure and vaccine escape are admittedly described quite naively in this work. I do not have objections to simplicity if put in perspective (as the authors do). However, I wonder whether it could be possible to translate the current definition of vaccine escape, which is not completely obvious to get dimensionally, to something like the probability of vaccine escape. I believe that this could be done quite easily (although perhaps at the expense of one additional parameter) if one defines $\text{Prob}(\text{vaccine escape})=1-\text{Prob}(\sim\text{vaccine escape})=1-(1-p)^{(C*P)}$, with p being the probability of vaccine escape within a single host.

- Following up on the previous point, the authors assume (in the main text) that only do infections in vaccinated people contribute to the risk of vaccine escape. However, they acknowledge that the situation is much more complicated in reality, and even relax their hypothesis (in the supplements) by accounting for the role possibly played by infection in unvaccinated people. As a matter of fact, every infection gives the virus new chances to evolve, by genetic drift if not by selection. With viral transmission still rampant and vaccine rollout still slow in many countries, understanding what mechanism contributes the most to evolutionary dynamics is of course challenging (leaving aside competition dynamics, which would require a more complex modeling framework). That is why it would seem important to me to include at least part of the section about the sensitivity analysis of vaccine escape results, along with Figure S2, in the main text.

- The manuscript is generally well written and quite easy to follow. However, there exist several instances where writing could be further improved for clarity. I am attaching a copy of the manuscript file (see Appendix A) with some minor remarks and suggestions marked in green (plus some notes of mine which have been translated into the comments above).

Review form: Reviewer 2

Is the manuscript scientifically sound in its present form?

Yes

Are the interpretations and conclusions justified by the results?

Yes

Is the language acceptable?

Yes

Do you have any ethical concerns with this paper?

No

Have you any concerns about statistical analyses in this paper?

No

Recommendation?

Accept as is

Comments to the Author(s)

The manuscript "Vaccine escape in a heterogeneous population: insights for SARS-CoV-2 from a simple model" by Gog et al. analyses a simple model for vaccination in a heterogeneous population, to infer some general principles, that may be useful for designing actual vaccination strategies.

In a stylized population consisting of two groups, one with a higher contact rate, the other one subject to more serious complication if infected, the authors study in which group it is more convenient allocating limited vaccine resources, according to different criteria.

The model is simple enough that analytical formulae can be obtained and computed to answer the question. The answer depends of course on parameter values and on the criterion used; the authors conclude anyway that “in the majority of the parameter space explored, vaccinating the mixers is more effective than vaccinating the vulnerable to reduce the total amount of disease”. This result, valid as long as vaccines are able to limit, at least partially, the transmission of the infection and there is a significant difference in contact rates between the two groups, is in line with the general epidemiological theory. I must however remark that, if we are thinking of COVID-19 and the groups represent different younger and older age classes, the value of the parameter d should be around 1,000 (see, e.g. O’Driscoll et al, 2021) rather than in the range 1-10, and this would make quite a difference. Possibly this is one of the reasons for the different result obtained in [44], beyond the ones offered by the authors. I think that the authors should at least acknowledge the issue.

The more novel part of the article concerns the effect of vaccination policy on the probability of vaccine escape. While the model is very simple and the results are difficult to interpret in terms of actual policies, it is important bringing the point to both modellers and public health authorities, and the general principle (intermediate vaccination rates maximize the risk) appears to be robust. I think that the manuscript is interesting and worthwhile. The authors recognize the limitation of the model used, and they discuss with competence whether their results are expected to be robust to model details.

In the Supplementary Material the authors show the effect of some changes in the model or in the parameter values used. I would have been interested in seeing the effect of at least two other modifications:

- the authors always assume proportional mixing among the two groups. What if mixing is to some degree assortative?
 - the model assumes that some part of the population is vaccinated at $t=0$, and then the epidemic proceeds exponentially according to the resulting parameter values. Would the picture be different if vaccinations occur dynamically? Namely, they occur at some prescribed rate during the time period analysed. I understand that the problem is much more complex, as there would be no simple formula to evaluate the output, and simulations would be required. Furthermore, the model could become more complex, as one may think that public health authorities relax NPIs as a larger fraction of the population becomes vaccinated, bringing economic issues in the optimization, as already suggested by the authors at page 15. Still, I think it is an issue that is worth being analysed in as simple a context as possible.
- If the authors find the time to briefly analyse these issues, I think it would be an interesting addition to the manuscript, but this is only a suggestion.

Reference cited

O’Driscoll, M., Ribeiro Dos Santos, G., Wang, L., Cummings, D. A. T., Azman, A. S., Paireau, J., Fontanet, A., Cauchemez, S., & Salje, H. (2021). Age-specific mortality and immunity patterns of SARS-CoV-2. *Nature*, 590(7844), 140–145. <https://doi.org/10.1038/s41586-020-2918-0>

Decision letter (RSOS-210530.R0)

Dear Professor Gog

On behalf of the Editors, we are pleased to inform you that your Manuscript RSOS-210530 "Vaccine escape in a heterogeneous population: insights for SARS-CoV-2 from a simple model" has been accepted for publication in Royal Society Open Science subject to minor revision in accordance with the referees' reports. Please find the referees' comments along with any feedback from the Editors below my signature.

Please submit your revised manuscript and required files (see below) no later than 7 days from today's (ie 14-Jun-2021) date. Note: the ScholarOne system will 'lock' if submission of the revision is attempted 7 or more days after the deadline. If you do not think you will be able to meet this deadline please contact the editorial office immediately.

on behalf of Professor Enrico Bertuzzo (Associate Editor) and Nick Pearce (Subject Editor)
openscience@royalsociety.org

Associate Editor Comments to Author (Professor Enrico Bertuzzo):

Associate Editor: 1

Comments to the Author:

Both reviewers found the manuscript interesting and sound but they also highlighted some areas of improvement, especially in the presentation. I welcome the authors to revise the manuscript according to these suggestions.

Reviewer comments to Author:

Reviewer: 1

Comments to the Author(s)

The manuscript by Dr. Gog and colleagues deals with the analysis of a SIR-like epidemiological model applied to the transmission of SARS-CoV-2. Using the model, the authors discuss several vaccination strategies for a population composed of subgroups characterized by different mixing and vulnerability patterns. The focus of the analysis, besides the derivation of standard epidemiological metrics such as the reproduction number and the incidence of infection, is on the possibility for vaccines to exert selection pressure on the virus, ultimately resulting in the emergence of mutations that may be able to escape the immune response triggered by the administration of the vaccine.

Needless to say, the topic is of extreme interest. The almost equation-free approach used by the authors may also serve well the purpose of widening the readership of an otherwise technical manuscript. The toy-like nature of the model seems to be better suited to seek general mechanisms rather than specific decision-making prescriptions. This point is effectively addressed in the manuscript and should not be seen, in my view, as a limitation of this study. The presented results seem sound, given the hypotheses laid out by the authors.

That being said, I have some technical comments that the authors may want to consider while revising their work:

- The complexity of the model analyzed in the main text is kept to a minimum---and, I would argue, understandably so. Of the several simplifying hypotheses that have been introduced, one leaves me a bit perplexed, though: namely, that the two groups have the same relative abundance within the population. Besides the obvious unlikelihood of such numerical coincidence, I wonder whether this choice could perhaps lead to an underestimation (not quite by the authors, rather by some readers) of possible asymmetries in the transmission process and in the definition of epidemiological patterns. I am especially referring to the analytical treatment, where the 'vulnerable'-to-'mixer' ratio is nowhere to be found, exactly because of this strong 1-to-1 hypothesis. However, this ratio influences several of the results presented in this work, as acknowledged (and even shown) by the authors. I would suggest removing this 1-to-1 hypothesis from the main text while keeping all the other simplifications in place. Numerically, I would not change anything, meaning that the main text could still just account for the case $\epsilon=1$ (borrowing from the extended model presented in the main text).

- I am not against the 'direct calculation' approach chosen by the authors for the definition of the next-generation matrix. However, equation (2) needs to be better framed and more explicitly explained to make sure that readers can easily follow. For instance, I believe that at least some future readers might be left somehow dumbfounded by the fact that the fraction of vaccinated infectious people does not appear in the last two entries of the first row of the next-generation matrix (similar remarks apply to other entries as well). A similar observation holds also for equations (3) and (4), which are introduced basically with no prior methodological background. I believe that in all these cases the authors would do the less mathematically-versed readers a solid if they could expand just a bit the explanation of these technical aspects of their work.

- Some epidemiological terms need to be better defined. For instance, I cannot fully understand what do the authors mean when they say that in their model "incidence $I(t)$ is exponential, with growth rate λ " (p.4, l.39). Now, if they have in mind a model like $dI/dt=\lambda*I$, then I guess that $I(t)$ would be the cumulative incidence at time t ; if so, I do not get where the integral in equation (5) comes from. Some further explanation seems to be warranted here. The same goes for the term "prevalence", which seems to be used naively (in both the abstract and the summary).

- Selection pressure and vaccine escape are admittedly described quite naively in this work. I do not have objections to simplicity if put in perspective (as the authors do). However, I wonder whether it could be possible to translate the current definition of vaccine escape, which is not completely obvious to get dimensionally, to something like the probability of vaccine escape. I believe that this could be done quite easily (although perhaps at the expense of one additional parameter) if one defines $\text{Prob}(\text{vaccine escape})=1-\text{Prob}(\sim\text{vaccine escape})=1-(1-p)^{(C*P)}$, with p being the probability of vaccine escape within a single host.

- Following up on the previous point, the authors assume (in the main text) that only do infections in vaccinated people contribute to the risk of vaccine escape. However, they

acknowledge that the situation is much more complicated in reality, and even relax their hypothesis (in the supplements) by accounting for the role possibly played by infection in unvaccinated people. As a matter of fact, every infection gives the virus new chances to evolve, by genetic drift if not by selection. With viral transmission still rampant and vaccine rollout still slow in many countries, understanding what mechanism contributes the most to evolutionary dynamics is of course challenging (leaving aside competition dynamics, which would require a more complex modeling framework). That is why it would seem important to me to include at least part of the section about the sensitivity analysis of vaccine escape results, along with Figure S2, in the main text.

- The manuscript is generally well written and quite easy to follow. However, there exist several instances where writing could be further improved for clarity. I am attaching a copy of the manuscript file with some minor remarks and suggestions marked in green (plus some notes of mine which have been translated into the comments above).

Reviewer: 2

Comments to the Author(s)

The manuscript "Vaccine escape in a heterogeneous population: insights for SARS-CoV-2 from a simple model" by Gog et al. analyses a simple model for vaccination in a heterogeneous population, to infer some general principles, that may be useful for designing actual vaccination strategies.

In a stylized population consisting of two groups, one with a higher contact rate, the other one subject to more serious complication if infected, the authors study in which group it is more convenient allocating limited vaccine resources, according to different criteria.

The model is simple enough that analytical formulae can be obtained and computed to answer the question. The answer depends of course on parameter values and on the criterion used; the authors conclude anyway that "in the majority of the parameter space explored, vaccinating the mixers is more effective than vaccinating the vulnerable to reduce the total amount of disease". This result, valid as long as vaccines are able to limit, at least partially, the transmission of the infection and there is a significant difference in contact rates between the two groups, is in line with the general epidemiological theory. I must however remark that, if we are thinking of COVID-19 and the groups represent different younger and older age classes, the value of the parameter d should be around 1,000 (see, e.g. O'Driscoll et al, 2021) rather than in the range 1-10, and this would make quite a difference. Possibly this is one of the reasons for the different result obtained in [44], beyond the ones offered by the authors. I think that the authors should at least acknowledge the issue.

The more novel part of the article concerns the effect of vaccination policy on the probability of vaccine escape. While the model is very simple and the results are difficult to interpret in terms of actual policies, it is important bringing the point to both modellers and public health authorities, and the general principle (intermediate vaccination rates maximize the risk) appears to be robust. I think that the manuscript is interesting and worthwhile. The authors recognize the limitation of the model used, and they discuss with competence whether their results are expected to be robust to model details.

In the Supplementary Material the authors show the effect of some changes in the model or in the parameter values used. I would have been interested in seeing the effect of at least two other modifications:

- the authors always assume proportional mixing among the two groups. What if mixing is to some degree assortative?
- the model assumes that some part of the population is vaccinated at $t=0$, and then the epidemic proceeds exponentially according to the resulting parameter values. Would the picture be different if vaccinations occur dynamically? Namely, they occur at some prescribed rate during the time period analysed. I understand that the problem is much more complex, as there would

be no simple formula to evaluate the output, and simulations would be required. Furthermore, the model could become more complex, as one may think that public health authorities relax NPIs as a larger fraction of the population becomes vaccinated, bringing economic issues in the optimization, as already suggested by the authors at page 15. Still, I think it is an issue that is worth being analysed in as simple a context as possible.

If the authors find the time to briefly analyse these issues, I think it would be an interesting addition to the manuscript, but this is only a suggestion.

Reference cited

O'Driscoll, M., Ribeiro Dos Santos, G., Wang, L., Cummings, D. A. T., Azman, A. S., Paireau, J., Fontanet, A., Cauchemez, S., & Salje, H. (2021). Age-specific mortality and immunity patterns of SARS-CoV-2. *Nature*, 590(7844), 140–145. <https://doi.org/10.1038/s41586-020-2918-0>

===PREPARING YOUR MANUSCRIPT===

===PREPARING YOUR REVISION IN SCHOLARONE===

Author's Response to Decision Letter for (RSOS-210530.R0)

See Appendix B.

Decision letter (RSOS-210530.R1)

Dear Professor Gog,

It is a pleasure to accept your manuscript entitled "Vaccine escape in a heterogeneous population: insights for SARS-CoV-2 from a simple model" in its current form for publication in Royal Society Open Science. The comments of the reviewer(s) who reviewed your manuscript are included at the foot of this letter.

COVID-19 rapid publication process:

We are taking steps to expedite the publication of research relevant to the pandemic. If you wish, you can opt to have your paper published as soon as it is ready, rather than waiting for it to be published the scheduled Wednesday.

This means your paper will not be included in the weekly media round-up which the Society sends to journalists ahead of publication. However, it will still appear in the COVID-19 Publishing Collection which journalists will be directed to each week (<https://royalsocietypublishing.org/topic/special-collections/novel-coronavirus-outbreak>).

If you wish to have your paper considered for immediate publication, or to discuss further, please notify openscience_proofs@royalsociety.org and press@royalsociety.org when you respond to this email.

on behalf of Professor Enrico Bertuzzo (Associate Editor) and Nick Pearce (Subject Editor)
openscience@royalsociety.org

Associate Editor Comments to Author (Professor Enrico Bertuzzo):

The authors have convincingly revised the manuscript following the referees- suggestions. Please note that something went wrong in the compilation of the latest version and all the references are missing. Make sure to fix this when preparing the final files.

Appendix A**ROYAL SOCIETY
OPEN SCIENCE****Vaccine escape in a heterogeneous population: insights for
SARS-CoV-2 from a simple model**

Journal:	Royal Society Open Science
Manuscript ID	RSOS-210530
Article Type:	Research
Date Submitted by the Author:	31-Mar-2021
Complete List of Authors:	Gog, Julia; University of Cambridge, Department of Applied Mathematics and Theoretical Physics Hill, Edward; University of Warwick, Zeeman Institute: Systems Biology and Infectious Disease Epidemiology Research (SBIDER) Danon, Leon; University of Bristol, Department of Engineering Mathematics Thompson, Robin; University of Warwick, Zeeman Institute: Systems Biology and Infectious Disease Epidemiology Research (SBIDER)
Subject:	health and disease and epidemiology < BIOLOGY, theoretical biology < BIOLOGY
Keywords:	SARS-CoV-2, COVID-19, Vaccine, Vaccine escape, heterogeneous population, policy
Subject Category:	Science, Society and Policy

Author-supplied statements

Relevant information will appear here if provided.

Ethics

Does your article include research that required ethical approval or permits?:

This article does not present research with ethical considerations

Statement (if applicable):

CUST_IF_YES_ETHICS :No data available.

Data

It is a condition of publication that data, code and materials supporting your paper are made publicly available. Does your paper present new data?:

My paper has no data

Statement (if applicable):

CUST_IF_YES_DATA :No data available.

Conflict of interest

I/We declare we have no competing interests

Statement (if applicable):

CUST_STATE_CONFLICT :No data available.

Authors' contributions

This paper has multiple authors and our individual contributions were as below

Statement (if applicable):

Conceptualization - JRG, EJH, LD, RT

Formal Analysis - JRG

Methodology - JRG, EJH, LD, RT

Software - JRG

Vizualisation - JRG

Writing " original draft - JRG, EJH

Writing " review & editing - JRG, EJH, LD, RT

Vaccine escape in a heterogeneous population: insights for SARS-CoV-2 from a simple model

Julia R. Gog^{*1,2}, Edward M. Hill^{2,3,4,5}, Leon Danon^{2,6,7}, and Robin Thompson^{2,3,4}

¹*Department of Applied Mathematics and Theoretical Physics, University of Cambridge*

²*JUNIPER – Joint UNiversities Pandemic and Epidemiological Research,*
<https://maths.org/juniper/>

³*The Zeeman Institute for Systems Biology & Infectious Disease Epidemiology Research,*
University of Warwick

⁴*Mathematics Institute, University of Warwick*

⁵*School of Life Sciences, University of Warwick*

⁶*Department of Engineering Mathematics, University of Bristol*

⁷*The Alan Turing Institute*

This version: March 28, 2021

Abstract

As a counter measure to the SARS-CoV-2 pandemic there has been swift development and clinical trial assessment of candidate vaccines, with subsequent deployment as part of mass vaccination campaigns. However, the SARS-CoV-2 virus has demonstrated the ability to mutate and develop variants, which can modify epidemiological properties and potentially also the effectiveness of vaccines.

The widespread deployment of highly effective vaccines may rapidly exert selection pressure on the SARS-CoV-2 virus directed towards mutations that escape the vaccine-induced immune response. This is particularly concerning whilst infection is widespread. By developing and analysing a mathematical model of two population groupings with differing vulnerability and contact rates, we explore the impact of the deployment of vaccine amongst the population on R , cases, disease abundance and vaccine escape pressure.

→ The results from this model illustrate two insights: (i) vaccination aimed at reducing prevalence could be more effective at reducing disease than directly vaccinating the vulnerable; (ii) the highest risk for vaccine escape can occur at intermediate levels of vaccination. This work demonstrates → a key principle that the careful targeting of vaccines towards particular population groups could reduce disease as much as possible whilst limiting the risk of vaccine escape.

←
not defined

1 Introduction

SARS-CoV-2 has caused a global pandemic with over 115,000,000 reported cases and 2,500,000 confirmed deaths as of 7th March 2021 [1]. In response, multiple vaccine candidates have been rapidly developed, tested in international trials and rolled out in mass vaccination campaigns in many parts of the world [2].

*corresponding author: jrg20@cam.ac.uk

In the United Kingdom, two ^{vaccines} ~~vaccinations~~ are in use (as of March 2021), an mRNA-based vaccine pro-
\rightarrow duced by Pfizer, and a viral ~~vector~~ ^(obvious) ~~coronavirus~~ vaccine produced by AstraZeneca. Phase ~~3~~ ³ trials \leftarrow
have determined these vaccines to be highly effective against disease, with the mRNA-based vac-
cines, in particular, reporting central efficacies against disease (i.e. preventing COVID-19 symptoms)
in the range of 94% to 95% [3, 4].

With SARS-CoV-2, there remains considerable virological, epidemiological and immunological uncer-
tainty, with implications for vaccine escape currently underdeveloped. In the absence of vaccination,
the SARS-CoV-2 virus has demonstrated the ability to mutate and develop variants [5]. Variants with
multiple genetic changes have led to phenotypic changes increasing transmissibility [6, 7], mortal-
ity [8] and have the potential to reduce the effectiveness of vaccines [5]. The mass deployment of
highly effective vaccines, whilst infection is widespread, may rapidly exert selection pressure on the
SARS-CoV-2 virus directed towards mutations that escape the vaccine ~~induced~~ ^{induced} immune response. \leftarrow
However, the strength of this selection and the likelihood of vaccine escape is unknown at this time [9].

Due to limited vaccine supply, countries must decide on priority orders for vaccination. The optimal
order of prioritisation will depend upon the measure being optimised (i.e. protecting essential societal
functions or directly minimising health harms, such as cases, hospitalisations or deaths, or some
combination of these) [10, 11, 12]. In the United Kingdom, vaccination policy advice is provided by
the Joint Committee on Vaccination and Immunisation (JCVI). The JCVI advised that the first priorities
for the SARS-CoV-2 vaccination programme should be the prevention of COVID-19 mortality and the
\Rightarrow protection of health ~~and social~~ ^{and social} care staff and systems [13]. At the time of the initial prioritisation,
extremely limited data were available from clinical trials on vaccine efficacy for preventing infection
and onward transmission. For the second phase of the vaccination programme, JCVI was asked
by the Department for Health and Social Care (DHSC) to formulate advice on the optimal strategy
to further reduce mortality, morbidity and hospitalisations from COVID-19 disease. The subsequent
advice given was to proceed with an age-based priority order, with operational considerations as part
of the justification on account of speed of vaccine uptake being paramount [14].

For prospective investigations, in the absence of empirical data, mathematical models provide a
method to gather insight on these questions. We explore the interactions between the deployment of
vaccine amongst the population, infection and disease prevalence, and vaccine escape. In this work,
we ask the question of how considerations of vaccine escape risk might modulate optimal vaccine
priority order. In particular, if infection in vaccinated individuals contributes to pressure to generate
vaccine escape, how do the risks depend on the parts of the population that have been vaccinated.
Rather than aiming to develop a detailed model of SARS-CoV-2 transmission dynamics, we present
a two-population model with differing vulnerability and contact rates to elucidate broad principles on
the relationships between epidemiological regimes, vaccine efficacy and vaccine escape. We explore
strategies without the constraint of matching the vaccination rollout that has already happened in any
country, both for applicability to future scenarios and to other countries.

2 Methods

2.1 Population heterogeneity

We are taking the approach of directly building the next generation matrix, based on assumptions
about the population and effects of vaccination. We capture population variability in vulnerability and
mixing by dividing our model population into two equally sized groups: half of the population are more
vulnerable to disease and mix less with others, the other half is less vulnerable but mixes more with
others – as shorthand we term these two halves of the population as ‘vulnerable’ and ‘mixers’. The

*this is quite a non-generic assumption:
I would remove it from the basic model (the NGM might even be easier to read with some more built-in is/symmetries)*

<https://royalsocietypublishing.org/journal/rsos>

assumption of equal proportions is taken for simplicity, but the effects of relaxing this assumption are explored in the Supplementary Information (Figures S8 and S9). Vulnerability is modelled simply as a ratio $d > 1$ of a higher chance of a severe outcome if a vulnerable individual is infected compared to if a mixer is infected. This might represent progression to hospitalisation, need for more intensive treatment or a higher mortality rate. In practice of course, all of these could be separate effects, and 'vulnerability' is not straightforward. However to gain broad insights here, vulnerability is treated in this simple way – a higher chance of poor outcome, termed 'disease' in the results below for brevity. For the more mixing (less vulnerable) half of the population, they are deemed to have an m times higher rate of contact with others than the rest of the population (all the rest being vulnerable in this model). Carrying this through to a mixing matrix, this would be that mixers have m^2 higher mixing within their own group than non-mixers have within theirs, and m times higher between groups. To isolate and examine the key factors here of host vulnerability and mixing, we assume that the vulnerable and mixers are equally susceptible to infection, and also equally infectious if infected (only modified by their contact patterns). We also make the assumption in our analysis that there is no prior immunity in this system.

Consider reducing the use of unqualified "this"

2.2 Effects of vaccination

For vaccination, we ignore any delay of effect of vaccination and multiple doses, but we do split the effect of the vaccination into three components. In this model, vaccination can (i) reduce the risk of infection, (ii) reduce the risk of severe disease and (iii) reduce the risk of infecting others, and we capture these as θ_S , θ_D and θ_I . These θ are all separate multiplicative effects on their corresponding rates, and hence $\theta = 0$ corresponds to the vaccine having complete/perfect prevention of infection, fully preventing disease given infection or being fully infectivity blocking and $\theta = 1$ means having no effect of the corresponding type. The θ here are comparable to $1 - VE$ of Halloran *et al.*[15].

Translating this framework to a general idea of disease blocking, this is the combined effect of reducing susceptibility and disease $\theta_S \times \theta_D$ gives the relative risk of disease for someone vaccinated compared to unvaccinated (so vaccine efficacy in terms of disease blocking would be $1 - \theta_S \theta_D$, while vaccine efficacy in terms of case prevention would be $1 - \theta_S$). For transmission blocking, it is the combination of susceptibility and infectiousness that matters: $\theta_S \times \theta_I$ gives the relative contribution of population transmission from someone vaccinated compared to unvaccinated. It might be tempting mathematically to combine these to reduce this system to two parameters for vaccination, but all three distinct processes are needed to explore the number of vaccinated who become infected, as we argue we should when considering vaccine escape.

2.3 Direct calculation

Without vaccination, the next generation matrix (NGM, the matrix that relates the number of infected individuals of each type between infection generations [16]) is proportional to the matrix \mathbf{M}_0 , given by:

$$\mathbf{M}_0 = \begin{bmatrix} 1 & m \\ m & m^2 \end{bmatrix} = \begin{bmatrix} 1 \\ m \end{bmatrix} \begin{bmatrix} 1 \\ m \end{bmatrix}^T \quad (1)$$

where the first population represents the vulnerable and the second the mixers. Suppose now that a proportion v_1 and v_2 of the vulnerable and the mixers have been vaccinated respectively. This population can now be thought of as split into four compartments: the two unvaccinated groups as before (unvaccinated vulnerable, unvaccinated mixers) and then the two corresponding vaccinated groups (vaccinated vulnerable, vaccinated mixers).

Now the NGM is four by four, and as the original NGM could be represented by an outer product, this vaccinated NGM is proportional to $M[v_1, v_2]$:

isn't the fraction of vaccinated vulnerable / mixers needed as well?

$$M[v_1, v_2] = \begin{bmatrix} (1-v_1) & (1-v_1)m & (1-v_1)\theta_I & (1-v_1)m\theta_I \\ (1-v_2)m & (1-v_2)m^2 & (1-v_2)m\theta_I & (1-v_2)m^2\theta_I \\ \theta_S v_1 & \theta_S v_1 m & \theta_S v_1 \theta_I & \theta_S v_1 m \theta_I \\ \theta_S v_2 m & \theta_S v_2 m^2 & \theta_S v_2 m \theta_I & \theta_S v_2 m^2 \theta_I \end{bmatrix} = \begin{bmatrix} (1-v_1) \\ (1-v_2)m \\ \theta_S v_1 \\ \theta_S v_2 m \end{bmatrix} \begin{bmatrix} 1 \\ m \\ \theta_I \\ m\theta_I \end{bmatrix}^T \quad (2)$$

When M can be written as an outer product, it is rank one and the spectral radius follows immediately (inner product of the same vectors, giving a positive real eigenvalue). The corresponding eigenvector can be read off (the column vector), giving the relative proportions of cases as split between the four groups. Further, under general feasible initial conditions (non-negative infections in all groups, perhaps zero in some but not all), the vector denoting the proportion of cases in each group will pivot quickly from any general initial distribution to this dominant eigenvector as all the other eigenvalues are zero.

among

The spectral radius (dominant eigenvalue here) of $M[v_1, v_2]$:

$$\sigma[v_1, v_2] = (1-v_1) + (1-v_2)m^2 + \theta_S \theta_I (v_1 + v_2 m^2) \quad (3)$$

where the transmission-blocking combination of vaccine parameters ($\theta_S \theta_I$) naturally emerges here. As the effective reproduction ratio is proportional to this σ , $R[v_1, v_2]$ is given by

Some derivation details would help, here

$$R[v_1, v_2] = R[0, 0] \frac{\sigma[v_1, v_2]}{\sigma[0, 0]} = R_0 \frac{(1-v_1) + (1-v_2)m^2 + \theta_S \theta_I (v_1 + v_2 m^2)}{1 + m^2}$$

and it is immediately apparent that that this it is linear in the proportions vaccinated.

We approximate the effective reproduction ratio as being constant during the period of time under consideration for assessing vaccine effects (t_{max}): in other words, there is no susceptible depletion as the timescale is relatively short in terms of the incidence under consideration (the lower the incidence, the longer this period can be). Then, the incidence $I(t)$ is exponential, with growth rate λ . Again for simplicity, we take $\lambda = \log(R)/T$ – the growth rate mapping from R corresponding to a fixed infectious period T with no variance. Then the incidence can be easily integrated over time to give the total number of cases during the period in question, and is further simplified by expressing the duration of the period of interest in terms of mean generation time T , so $t_{max} = GT$, where G is the duration of the period in terms of disease generations. We will consider the relative number of cases below, meaning constants unaffected by changing vaccination can be scaled out. We choose here to scale out initial incidence I_0 and also scale by t_{max} (to give $F(R)$ as something that could be interpreted as a time average of cases relative to initial incidence):

unclear, definition needed

$I = \lambda I$
↑
incidence
 $I(t) = I_0 e^{\lambda t}$
≡ cumul. incidence

$$F(R) = \frac{\int_0^{t_{max}} I(t) dt}{I_0 t_{max}} = \frac{\int_0^{t_{max}} I_0 e^{\lambda t} dt}{I_0 t_{max}} = \frac{e^{\lambda t_{max}} - 1}{\lambda t_{max}} = \frac{R^G - 1}{\log(R^G)} \quad (5)$$

for $R \neq 1$. Also, $F(1) = 1$ (either by L'Hôpital's Rule or the integral using $\lambda = 0$). From above, we then have the relative number of cases $C[v_1, v_2]$, compared to a scenario with no vaccination:

why the integral?

$$C[v_1, v_2] = \frac{F(R[v_1, v_2])}{F(R_0)} \quad (6)$$

and these cases are distributed in the four subpopulations in proportion to the dominant eigenvector from above (ordered unvaccinated vulnerable, unvaccinated mixers, vaccinated vulnerable, vaccinated mixers respectively in the vector), normalised to give proportion of cases which are in each group:

$$\mathbf{P}[v_1, v_2] = \frac{1}{(1 - v_1) + (1 - v_2)m + \theta_S v_1 + \theta_S v_2 m} \begin{bmatrix} (1 - v_1) \\ (1 - v_2)m \\ \theta_S v_1 \\ \theta_S v_2 m \end{bmatrix} \quad (7)$$

2.4 Output metrics

We consider four main outputs. Two are already established above: the effective reproduction rate ($R[v_1, v_2]$) and the relative number of cases ($C[v_1, v_2]$). We define a further two in this section: a measure of the amount of disease relative to no vaccination ($D[v_1, v_2]$) and a measure of vaccine escape pressure ($V[v_1, v_2]$).

For ‘disease’, we consider the severe outcomes as represented by the vulnerability parameter d (which could represent hospitalisation, mortality, or any proxy of interest for severity). We already have the relative number of cases (C , equation 6) and know how these are distributed among the four population groups (\mathbf{P} , equation 7). The relative risk of disease is multiplied by a factor of d for the vulnerable and θ_D for the vaccinated (and multiplied by both for the vaccinated vulnerable). For the four respective groups, ordered as previously, the relative risk of disease is proportional to \mathbf{U} :

$$\mathbf{U} = \begin{bmatrix} d \\ 1 \\ d\theta_D \\ \theta_D \end{bmatrix} \quad (8)$$

Combining these, we have $D[v_1, v_2]$: a measure of total disease relative to a scenario with no vaccination:

$$D[v_1, v_2] = C[v_1, v_2] \frac{\mathbf{P}[v_1, v_2] \mathbf{U}}{\mathbf{P}[0, 0] \mathbf{U}} \quad (9)$$

$$= C[v_1, v_2] \left(\frac{d(1 - v_1) + (1 - v_2)m + d\theta_D \theta_S v_1 + \theta_D \theta_S v_2 m}{(1 - v_1) + (1 - v_2)m + \theta_S v_1 + \theta_S v_2 m} \right) / \left(\frac{d + m}{1 + m} \right) \quad (10)$$

For ‘vaccine escape’, reality is a highly complex picture of variants being generated and selected at various scales within and between host [17, 18]. Here we take an extremely simple approach and measure pressure on vaccine escape as proportional to the number of cases in vaccinated individuals, treating the vulnerable and mixers as equal in this respect (sensitivity to including cases in unvaccinated hosts as contributing to the vaccine escape pressure is also considered below - see the Supplementary Information, Figure S2). It is far from clear that this is the best way to approach this, but we propose it here as a straightforward and achievable method. We acknowledge the shortcomings of this approach must be held in mind when interpreting the results below.

Building this mathematically, vaccine escape $V[v_1, v_2]$ is proportional to the number of cases in vaccinated individuals, but the normalisation for this cannot be the same quantity under no vaccination (this would be a zero denominator), so we use *total* number of cases under no vaccination as the normalisation. Let $P[v_1, v_2]$ be the proportion of cases that are in vaccinated individuals:

Can't this be translated into some probability via simple assumptions? e.g. 5 Prob (vac. esc.) = 1 - Prob (no vac. esc.) = 1 - (1-p) CP

unspecified

<https://mc.manuscriptcentral.com/rsos>

$$P[v_1, v_2] = \mathbf{P}[v_1, v_2] \cdot \begin{bmatrix} 0 \\ 0 \\ 1 \\ 1 \end{bmatrix} \quad (11)$$

Then $V[v_1, v_2]$ is the product of the relative cases ($C[v_1, v_2]$) and the proportion of these cases that are in vaccinated individuals ($P[v_1, v_2]$):

$$V[v_1, v_2] = C[v_1, v_2] P[v_1, v_2] \quad (12)$$

$$= C[v_1, v_2] \frac{\theta_S v_1 + \theta_S v_2 m}{(1 - v_1) + (1 - v_2)m + \theta_S v_1 + \theta_S v_2 m} \quad (13)$$

2.5 Extension to more general population structures

It is straightforward to generalise this to n population groups, where group i has relative size x_i of the population, a relative vulnerability d_i and relative mixing m_i (with one degree of freedom in each of these, so either one group can be set to unity, or total normalised). When considering more general population structures, relative susceptibility to disease or infectiousness to others can also be included (μ_i and τ_i respectively) – this may be particularly important if the population is broken down into age classes considering children separately.

Following analogously from above the next generation matrix is $2n \times 2n$ and can be written as an outer product:

$$NGM = \begin{bmatrix} (1 - v_1)x_1\mu_1m_1 \\ (1 - v_2)x_2\mu_2m_2 \\ \vdots \\ (1 - v_n)x_n\mu_nm_n \\ \theta_S v_1 x_1 \mu_1 m_1 \\ \theta_S v_2 x_2 \mu_2 m_2 \\ \vdots \\ \theta_S v_n x_n \mu_n m_n \end{bmatrix} \begin{bmatrix} m_1 \tau_1 \\ m_2 \tau_2 \\ \vdots \\ m_n \tau_n \\ \theta_I m_1 \tau_1 \\ \theta_I m_2 \tau_2 \\ \vdots \\ \theta_I m_n \tau_n \end{bmatrix}^T \quad (14)$$

As before, this is a rank one matrix and the spectral radius here is the inner product of the same vectors, giving the proportionality with the effective reproduction ratio R . The calculation for cases is exactly as above, and the distribution of cases is as the dominant eigenvector, which is the column vector of the outer product.

Further generalisations are implementable, for example the vaccine effects θ_S , θ_I and θ_D could vary by age group – this would require additional parameterisation but the same analytic approach remains possible. In the more general case that the mixing structure cannot be written as an outer product then it is likely a numerical approach would be needed.

2.6 Parameterisation

[revised manuscript text omitted]

For the vulnerability ratio d this is not straightforward to parameterise as (a) we are using this to explore severe outcomes in an abstract way, so it could correspond to probability of hospitalisation or

a case fatality ratio or any other measure of severe disease and (b) the simple two-population structure is for exploration of the effects of heterogeneity rather than explicitly corresponding to defined population groups. Further, estimates for COVID-19 severity vary between studies, depending on context [29, 30, 31] and presence of more pathogenic variants [8, 6]. Below, we have taken $d = 10$ as the default in plots to explore the case where the vulnerable group is substantially more at risk, but the other half of the populations cannot be neglected for disease risk. For results on disease below, these are shown for a range of d (1 to 10) and it is visually clear what would happen for larger d . Most of the results below on vaccine escape do not depend on d .

For the parameters for the scenario under consideration, we have considered a situation where $R > 1$ initially before vaccination, choosing particularly $R_0 = 1.2$ which approximately corresponds to mid-September and October 2020 in United Kingdom [32], a situation with some regions under tight restrictions and some interventions everywhere (this is clearly not a true R_0 , but here R_0 is termed for the value of the effective reproduction ratio at this time if there were no vaccination). The value of G , the time period considered as measured in mean generation times, is going to be a subjective decision. Estimates for the generation time are variable between studies, but typically around 4-6 days [33, 34]. We take $G = 15$ by default, corresponding to a time window of 2-3 months. How results vary with G is discussed below, and $G = 5$ is used example to show how outputs change with a shorter G in the Supplementary Information (Figures S3-S6).

3 Results

3.1 Dependency of epidemiological outcomes on vaccine coverage

A summary set of results for a typical parameter set are shown in Figure 1. The effective reproduction ratio decreases as more people are vaccinated (Figure 1 top left). From the analytic expression above, we can see that this decrease in the effective reproduction ratio occurs for all parameter values so long as there is any transmission-blocking effect of the vaccine ($\theta_S \theta_I < 1$). Further, the dependence on the proportion vaccinated is linear, with stronger effect (by factor m^2 here) for vaccinating the mixers. The cases (Figure 1 top right) are here a direct function of R so also decreases with increasing vaccination, but not linearly: there is a steep drop to $R = 1$ and there after the effect is smaller, simply reflecting prevalence dropping faster during the period in question. Intuitively, we expect similar vaccine effects on R and total cases will hold in more complex models.

3.2 Effect of vaccine on number of cases with severe disease

The total number of severe infections over this fixed period, denoted here as disease (Figure 1 bottom left), decreases as vaccination increases in either group. However this is no longer purely a function of R : it is also dependent on *who* is infected – the distribution of cases among the vulnerable and mixers. If vaccination coverage is higher in the vulnerable than the mixers, disease is disproportionately brought down relative to cases, and this is visible as a slight curve of the contours in the bottom right of the panel (where v_1 is high and v_2 is low).

It is intuitive that for a very wide range of models, vaccinating more people in any group has the effect of decreasing cases in that group and also possibly other groups also, driven by the dual effects of vaccination in transmission-blocking and disease-blocking effects. The question remains of which group it would be most effective to vaccinate to reduce severe disease (or any other outcome represented by increased vulnerability).

This question of optimal allocation can be explored by considering a situation in which we have a fixed

Figure 1: Summary outputs

Summary outputs for a fixed set of population parameters ($m = 2$, $d = 10$), vaccine parameters ($\theta_S = 0.6$, $\theta_T = 0.6$, $\theta_D = 0.3$) and scenario under consideration ($R_0 = 1.2$, $G = 15$). Four output measures are shown, relative reproduction ratio $R[v_1, v_2]$ (upper left), relative cases $C[v_1, v_2]$ (upper right), relative disease $D[v_1, v_2]$ (lower left), and vaccine escape pressure $V[v_1, v_2]$ (lower right). All four panels are shown as contour plots with horizontal and vertical axes representing the proportion of vulnerable and mixers vaccinated (v_1 and v_2 respectively).

Figure 2: Optimal allocation for averting disease

[revised manuscript text omitted]

1
2
3
4 → and mixers. In all of these, the total cases decreases with more vaccination (Figure 3 bottom middle panel), but the proportion of these cases which are in those who have been vaccinated increases with more vaccination (Figure 3 bottom left panel). The product of these gives the vaccine escape pressure, and for all of these it is unimodal: there is highest risk at some intermediate range of vaccination. This peak is maximised by vaccinating vulnerable first, but it is there for all paths for the parameters used here. *unpenalized*

11 These effects are dependent on the vaccine changing the trajectory of the epidemic and bringing cases down. For a shorter time horizon, there is less time for these effects to come into play. Similarly, if the cases in unvaccinated individuals played a significant role in vaccine escape, then this picture would be modified, mainly to reduce the low pressure for low vaccination. 
[revised manuscript text omitted]

- [3] Fernando P Polack, Stephen J Thomas, Nicholas Kitchin, Judith Absalon, Alejandra Gurtman, Stephen Lockhart, John L Perez, Gonzalo Pérez Marc, Edson D Moreira, Cristiano Zerbini, et al. Safety and efficacy of the bnt162b2 mrna covid-19 vaccine. *New England Journal of Medicine*, 383(27):2603–2615, 2020.
- [4] Lindsey R Baden, Hana M El Sahly, Brandon Essink, Karen Kotloff, Sharon Frey, Rick Novak, David Diemert, Stephen A Spector, Nadine Rouphael, C Buddy Creech, et al. Efficacy and safety of the mrna-1273 sars-cov-2 vaccine. *New England Journal of Medicine*, 384(5):403–416, 2021.
- [5] Public Health England. Investigation of SARS-CoV-2 variants of concern in England: Technical briefing 6. https://assets.publishing.service.gov.uk/government/uploads/system/uploads/attachment_data/file/961299/Variants_of_Concern_VOC_Technical_Briefing_6_England-1.pdf, 2021. Accessed: 12 March 2021.
- [6] Nicholas G Davies, Sam Abbott, Rosanna C Barnard, Christopher I Jarvis, Adam J Kucharski, James Munday, Carl AB Pearson, Timothy W Russell, Damien C Tully, Alex D Washburne, et al. Estimated transmissibility and severity of novel sars-cov-2 variant of concern 202012/01 in england. *MedRxiv*, 2021.
- [7] Erik Volz, Swapnil Mishra, Meera Chand, Jeffrey C Barrett, Robert Johnson, Lily Geidelberg, Wes R Hinsley, Daniel J Laydon, Gavin Dabrera, Áine O’Toole, et al. Transmission of sars-cov-2 lineage b. 1.1.7 in england: Insights from linking epidemiological and genetic data. *medRxiv*, 2021.
- [8] Robert Challen, Ellen Brooks-Pollock, Jonathan M Read, Louise Dyson, Krasimira Tsaneva-Atanasova, and Leon Danon. Increased hazard of mortality in cases compatible with sars-cov-2 variant of concern 202012/1-a matched cohort study. *British Medical Journal*, 372:n579, 2021.
- [9] Peter Horby, Muge Cevik, Angela McLean, Andrew Rambaut, Peter Openshaw, Meera Chand, Jake Dunning, and Wendy Barclay. Sars-cov-2 immune escape variants. <https://www.gov.uk/government/publications/sars-cov-2-immunity-escape-variants-7-january-2021>, 2021. Accessed: 8 March 2021.
- [10] Laura Matrajt, Julia Eaton, Tiffany Leung, Dobromir Dimitrov, Joshua T Schiffer, David A Swan, and Holly Janes. Optimizing vaccine allocation for COVID-19 vaccines: critical role of single-dose vaccination. *medRxiv*, page 2020.12.31.20249099, 2021.
- [11] Nicola Mulberry, Paul F Tupper, Christopher MacCabe, Erin Kirwin, and Caroline Colijn. Vaccine Rollout Strategies: The Case for Vaccinating Essential Workers Early. *medRxiv*, 2021.
- [12] Wei Wang, Qianhui Wu, Juan Yang, Kaige Dong, Xinghui Chen, Xufang Bai, Xinhua Chen, Zhiyuan Chen, Cécile Viboud, Marco Ajelli, et al. Global, regional, and national estimates of target population sizes for covid-19 vaccination: descriptive study. *bmj*, 371, 2020.

- 1
2
3
4 [13] Department of Health & Social Care. Joint Committee on Vaccination and Immunisation: advice on priority groups for COVID-19 vaccination, 30 December 2020. <https://www.gov.uk/government/publications/priority-groups-for-coronavirus-covid-19-vaccination-advice-from-the-jcvi-30-december-2020/joint-committee-on-vaccination-and-immunisation-advice-on-priority-groups-for-covid-19-vaccination-advice-from-the-jcvi-30-december-2020>. Accessed: 12 March 2021.
- 11 [14] Department of Health & Social Care. JCVI interim statement on phase 2 of the COVID-19 vaccination programme. <https://www.gov.uk/government/publications/priority-groups-for-phase-2-of-the-coronavirus-covid-19-vaccination-programme-advice-from-the-jcvi-interim-statement-on-phase-2-of-the-covid-19-vaccination-programme>, 2021. Accessed: 12 March 2021.
- 18 [15] M Elizabeth Halloran, Michael Haber, Ira M Longini Jr, and Claudio J Struchiner. Direct and indirect effects in vaccine efficacy and effectiveness. *American journal of epidemiology*, 133(4):323–331, 1991.
- 22 [16] Odo Diekmann, Johan Andre Peter Heesterbeek, and Johan AJ Metz. On the definition and the computation of the basic reproduction ratio r_0 in models for infectious diseases in heterogeneous populations. *Journal of mathematical biology*, 28(4):365–382, 1990.
- 26 [17] Sarah Cobey, Daniel B Larremore, Yonatan H. Grad, and Marc Lipsitch. Concerns about SARS-CoV-2 evolution should not hold back efforts to expand vaccination. *Pre-print, Harvard University*, 2021.
- 30 [18] Thompson R N, Hill E M, and Gog J R. Sars-cov-2 incidence and vaccine escape. *Lancet Infectious Disease*, 2021.
- 33 [19] Public Health England. Annex A: Report to JCVI on estimated efficacy of a single dose of Pfizer BioNTech (BNT162b2 mRNA) vaccine and of a single dose of ChAdOx1 vaccine (AZD1222). https://assets.publishing.service.gov.uk/government/uploads/system/uploads/attachment_data/file/949505/annex-a-phe-report-to-jcvi-on-estimated-efficacy-of-single-vaccine-dose.pdf, 2021. Accessed: 12 March 2021.
- 40 [20] Merryn Voysey, Sue Ann Costa Clemens, Shabir A Madhi, Lily Y Weckx, Pedro M Folegatti, Parvinder K Aley, Brian Angus, Vicky L Baillie, Shaun L Barnabas, Qasim E Bhorat, et al. Single-dose administration and the influence of the timing of the booster dose on immunogenicity and efficacy of chadox1 nCoV-19 (azd1222) vaccine: a pooled analysis of four randomised trials. *The Lancet*, 2021.
- 46 [21] Jamie Lopez Bernal, Nick Andrews, Charlotte Gower, Julia Stowe, Chris Robertson, Elise Tessier, Ruth Simmons, Simon Cottrell, Richard Robertson, Mark O’Doherty, Kevin Brown, Claire Cameron, Diane Stockton, Jim McMenamin, and Mary Ramsay. Early effectiveness of COVID-19 vaccination with BNT162b2 mRNA vaccine and ChAdOx1 adenovirus vector vaccine on symptomatic disease, hospitalisations and mortality in older adults in England. *medRxiv*, 2021.
- 54 [22] Catherine Hyams, Robin Marlow, Zandile Maseko, Jade King, Lana Ward, Kazminder Fox, Robyn Heath, Anabella Turner, Zsolt Friedrich, Leigh Morrison, Gabriella Ruffino, Rupert Antico, David Adegbite, Zsuzsa Szasz-Benczur, Maria Garcia Gonzalez, Jennifer Oliver, Leon Danon, and Adam Finn. Assessing the effectiveness of bnt162b2 and chadox1ncov-19 covid-19 vaccination in prevention of hospitalisations in elderly and frail adults: A single centre test negative case-control study. *SSRN*, 2021.

[23] Office for National Statistics. Coronavirus (COVID-19) weekly insights: latest health indicators in England, 12 February 2021. <https://www.ons.gov.uk/peoplepopulationandcommunity/healthandsocialcare/conditionsanddiseases/articles/coronaviruscovid19weeklyinsights/latesthealthindicatorsinengland12february2021#infections-hospital-admissions-and-deaths>, 2021. Accessed: 12 March 2021.
[24] Public Health England. Analysis of the relationship between pre-existing health conditions, ethnicity and COVID-19. https://assets.publishing.service.gov.uk/government/uploads/system/uploads/attachment_data/file/942091/Summary_report_ethnicity_and_comorbidity.pdf, 2021. Accessed: 12 March 2021.
[25] Yuanyuan Dong, Xi Mo, Yabin Hu, Xin Qi, Fang Jiang, Zhongyi Jiang, Shilu Tong, et al. Epidemiological characteristics of 2143 pediatric patients with 2019 coronavirus disease in china. *Pediatrics*, 145(6):e20200702, 2020.
[26] Xiaoxia Lu, Liqiong Zhang, Hui Du, Jingjing Zhang, Yuan Y Li, Jingyu Qu, Wenxin Zhang, Youjie Wang, Shuangshuang Bao, Ying Li, et al. Sars-cov-2 infection in children. *New England Journal of Medicine*, 382(17):1663–1665, 2020.
[27] Russell M Viner, Oliver T Mytton, Chris Bonell, GJ Melendez-Torres, Joseph Ward, Lee Hudson, Claire Waddington, James Thomas, Simon Russell, Fiona Van Der Klis, et al. Susceptibility to sars-cov-2 infection among children and adolescents compared with adults: a systematic review and meta-analysis. *JAMA pediatrics*, 2020.
[28] Petra Klepac, Adam J Kucharski, Andrew JK Conlan, Stephen Kissler, Maria L Tang, Hannah Fry, and Julia R Gog. Contacts in context: large-scale setting-specific social mixing matrices from the bbc pandemic project. *medRxiv*, 2020.
[29] Robert Verity, Lucy C Okell, Ilaria Dorigatti, Peter Winskill, Charles Whittaker, Natsuko Imai, Gina Cuomo-Dannenburg, Hayley Thompson, Patrick GT Walker, Han Fu, et al. Estimates of the severity of coronavirus disease 2019: a model-based analysis. *The Lancet infectious diseases*, 20(6):669–677, 2020.
[30] Graziano Onder, Giovanni Rezza, and Silvio Brusaferro. Case-fatality rate and characteristics of patients dying in relation to covid-19 in italy. *Jama*, 323(18):1775–1776, 2020.
[31] Ryosuke Omori, Kenji Mizumoto, and Gerardo Chowell. Changes in testing rates could mask the novel coronavirus disease (covid-19) growth rate. *International Journal of Infectious Diseases*, 94:116–118, 2020.
[32] UK Government. The reproduction number (R) and growth rate. https://assets.publishing.service.gov.uk/government/uploads/system/uploads/attachment_data/file/965192/R-and-growth-rate-time-series-26-Feb-2021.ods, 2021. Accessed: 12 March 2021.
[33] Luca Ferretti, Chris Wymant, Michelle Kendall, Lele Zhao, Anel Nurtay, Lucie Abeler-Dörner, Michael Parker, David Bonsall, and Christophe Fraser. Quantifying sars-cov-2 transmission suggests epidemic control with digital contact tracing. *Science*, 368(6491), 2020.
[34] John Griffin, Miriam Casey, Áine Collins, Kevin Hunt, David McEvoy, Andrew Byrne, Conor McAloon, Ann Barber, Elizabeth Ann Lane, and Simon More. Rapid review of available evidence on the serial interval and generation time of covid-19. *BMJ open*, 10(11):e040263, 2020.
[35] Julia R Gog, Lorenzo Pellis, James LN Wood, Angela R McLean, Nimalan Arinaminpathy, and James O Lloyd-Smith. Seven challenges in modeling pathogen dynamics within-host and across scales. *Epidemics*, 10:45–48, 2015.

[36] Simon Rella, Yuliya Kulikova, Emmanouil Dermitzakis, and Fyodor Kondrashov. Sars-cov-2
transmission, vaccination rate and the fate of resistant strains. *medRxiv*, 2021.
[37] Stephen M Kissler, Christine Tedijanto, Edward Goldstein, Yonatan H Grad, and Marc Lipsitch.
Projecting the transmission dynamics of sars-cov-2 through the postpandemic period. *Science*,
368(6493):860–868, 2020.
[38] Chadi M Saad-Roy, Caroline E Wagner, Rachel E Baker, Sinead E Morris, Jeremy Farrar, An-
drea L Graham, Simon A Levin, Michael J Mina, C Jessica E Metcalf, and Bryan T Grenfell.
Immune life history, vaccination, and the dynamics of sars-cov-2 over the next 5 years. *Science*,
370(6518):811–818, 2020.
[39] Chadi M Saad-Roy, Sinead E Morris, C Jessica E Metcalf, Michael J Mina, Rachel E Baker,
Jeremy Farrar, Edward C Holmes, Oliver Pybus, Andrea L Graham, Simon A Levin, et al. Epi-
demiological and evolutionary considerations of sars-cov-2 vaccine dosing regimes. *medRxiv*,
2021.
[40] Jennie S Lavine, Ottar N Bjornstad, and Rustom Antia. Immunological characteristics govern
the transition of covid-19 to endemicity. *Science*, 371(6530):741–745, 2021.
[41] Bryan T Grenfell, Oliver G Pybus, Julia R Gog, James LN Wood, Janet M Daly, Jenny A Mumford,
and Edward C Holmes. Unifying the epidemiological and evolutionary dynamics of pathogens.
*science*, 303(5656):327–332, 2004.
[42] Bina Choi, Manish C Choudhary, James Regan, Jeffrey A Sparks, Robert F Padera, Xueting
Qiu, Isaac H Solomon, Hsiao-Hsuan Kuo, Julie Boucau, Kathryn Bowman, et al. Persistence
and evolution of sars-cov-2 in an immunocompromised host. *New England Journal of Medicine*,
383(23):2291–2293, 2020.
[43] H el ene Chabas, S ebastien Lion, Antoine Nicot, Sean Meaden, Stineke van Houte, Sylvain
Moineau, Lindi M Wahl, Edze R Westra, and Sylvain Gandon. Evolutionary emergence of infec-
tious diseases in heterogeneous host populations. *PLoS biology*, 16(9):e2006738, 2018.
[44] Sam Moore, Edward M Hill, Louise Dyson, Michael Tildesley, and Matt J Keeling. Modelling
optimal vaccination strategy for sars-cov-2 in the uk. *medRxiv*, 2020.
[45] Sam Moore, Edward M Hill, Michael J Tildesley, Louise Dyson, and Matt J Keeling. Vaccination
and non-pharmaceutical interventions: When can the uk relax about covid-19? *medRxiv*, 2021.

5 Supplementary Information

5.1 Sensitivity of vaccine escape results

I think that this section could go to the main text if space allows

Figure S1 is analogous to the bottom right panel of Figure 1, but exploring a range of different transmission-blocking parameters for the effect of vaccination. Essentially the same qualitative effect is visible except when the vaccine has no transmission-blocking effects ($\theta_S = \theta_I = 1$, top left in Figure S1). In this case, increasing vaccination will not alter the number of cases going forward, and the only effect in terms of vaccine escape is to increase the number of cases which are in vaccinated individuals.

Apart from when there is little or no transmission-blocking, the maximum pressure on vaccine escape is exerted for $v_1 = 1, v_2 = 0$: in other words, vaccinating all of the vulnerable and none of the mixers. Even with all of the vulnerable vaccinated, the effective reproduction ratio and thus total cases are held high by the core of transmission within the mixing group. This transmission spills into the vulnerable vaccinated as the vaccine is not fully blocking infection ($\theta_S > 0$), thus ensuring a continued significant number of infections in the vaccinated, providing the platform for vaccine escape pressure. This effect will disappear if $\theta_S = 0$ – for a vaccine with perfect prevention of infection there would be no cases amongst the vaccinated.

Our simple measure of vaccine escape pressure is directly proportional to the number of cases in vaccinated individuals. This strict assumption can be related by supposing that cases in unvaccinated individuals also contribute, but at some lower level (Figure S2). So long as the unvaccinated cases do not contribute much (around $< 10\%$ as much as vaccinated for these parameter values), then the picture is qualitatively unchanged. However if unvaccinated cases do contribute more significantly, then by force of numbers, the picture is changed, specifically vaccine escape pressure is not low for little or no vaccination. In Figure S2 the bottom left of the panels (corresponding to low v_1 and v_2) changes the most as the weight of unvaccinated contribution to escape is increased, going through the panels.

If the contribution of unvaccinated to escape pressure is larger still, vaccine pressure will simply correspond more closely to total cases. In this case, vaccine escape pressure will be most quickly reduced by vaccinating the mixers first, corresponding with results on minimising disease.

5.2 Effect of a short time horizon

Results in the main text are given for $G = 15$ which corresponds to choosing a time horizon of 15 generation times of infection. Some of the dynamics above are underpinned by vaccination pushing down the number of cases over this period. This effect will be less marked if instead our focus is on a shorter time interval, when vaccination has not had time to accumulate its impacts on the epidemic trajectory. Equivalent plots to the main text are shown here for $G = 5$ in Figures S3, S4 and S5 and the equivalent to Figures S1 and S2 are in Figures S6 and S7.

In Figure S3, the qualitative results are similar to before: vaccination universally reduces R, cases and disease, and vaccination escape is similar except the maximum is now achieved by vaccinating all the vulnerables and some of the mixers.

Figure S4 shows that there is a wider parameter range now where it is optimal to vaccinate the vulnerable before the mixers to reduce disease. This shift fits with the balance between direct effects of protection against disease and longer effects of reshaping the epidemic: the shorter focus with $G = 5$ means the former dominates for more of the parameter range. However, even here it remains optimal to vaccinate mixers to reduce disease so long as there significant transmission-blocking effects and

there is heterogeneity in mixing (e.g. $m = 2$ here).

Figure S5 shows the same monotonicity for the two factors that make up vaccine escape pressure: total cases and the proportion of these cases that are in vaccinateds. Here, however, again the change in balance of effects with the shorter G means that vaccine escape pressure is not always maximal at intermediate vaccination (e.g. for the purple and blue paths in bottom right panel). However, these effects will be restored for stronger transmission-blocking assumptions (see Figure S6) - by the bottom right ($\theta_S = \theta_I = 0.4$) any route to full vaccination must pass a phase of higher vaccine escape pressure.

Figure S7 combines exploring sensitivity to the assumption that unvaccinated individuals can contribute to vaccine escape with the shorter time horizon $G = 5$. Interestingly, the combination of the two effects again can restore the picture of maximal vaccine escape pressure when all of the vulnerable and none of the mixers are vaccinated.

5.3 Relaxing assumption of equal-sized populations

Figures S8 and S9 explore breaking the assumption that the vulnerable and mixer populations are of equal size. We use the methods for the extension to population structure, though we retain two populations ($n = 2$). The relative size of the proportion of the vulnerable is given by x (so $x_1 = x$ and $x_2 = 1 - x$, say). In both Figures S8 and S9, the rows correspond to $x = 2/8, 4/6, 6/4, 8/2$, corresponding to the vulnerable being 20%, 40%, 60%, 80% of the population respectively. It should be borne in mind that when the two groups are not equally sized, the effort to vaccinate proportions of each group (v_1 and v_2) are not so directly comparable. For the ratio of disease averted (W in main text), the proportion of either group vaccinated (ϵ in main text) is adjusted to be equal absolute size as x is varied.

Figure S8 shows that the results do not vary qualitatively as the proportions are varied, except for a large proportion of vulnerable, the maximal vaccine escape pressure moves from vaccinating all of the vulnerable to vaccinating only some of them. The range where allocating a fixed small amount of vaccine to the vulnerable is optimal shrinks when vulnerable are a larger proportion of the population (Figure S8 bottom right) and grows when they are a small proportion (Figure S8 top right).

However, we are concerned that varying proportions of vulnerable and mixers might not be comparing like with like: Figure S8 keeps $d = 10$ for the vulnerable group and $m = 2$ for the mixers. An alternative would be to adjust these so as to concentrate or dilute vulnerability and mixing as the group sizes changed. We investigate this in S9. As x is varied, we also vary the vulnerability and mixing parameters to in effect keep a nominal excess mixing or vulnerability concentrated according to population sizes. We take $d_2 = 1$ still and $d_1 = 1 + \hat{d}/x$, so there is a baseline relative vulnerability of 1, and the excess of \hat{d} is shared between the vulnerable group of size x . Similarly with mixing: $m_1 = 1$, $m_2 = 1 + x\hat{m}$ so the extra mixing is shared among the mixing group which has relative size $1/x$. For the ratio of disease averted plots, the ranges of d and m are correspondingly varied. Setting $\hat{d} = 9$ and $\hat{m} = 1$, the default parameter set is recovered at $x = 1$.

Figure S9 shows that this adjustment still means that R , cases, disease and vaccine escape pressure do not vary much qualitatively. However now as plots for R , cases and disease against v_1 and v_2 they are also very similar quantitatively: this adjustment of d and m as functions of x keeps the plots almost invariant. The plot for vaccine escape pressure keeps the same overall shape, peaking with all the vulnerable vaccinated and none of the mixers. The ratio of disease averted is now sensitive to changing the proportion split, particularly at extremes. When all of the vulnerability is concentrated into a small proportion (Figure S9 top right panel) then vaccinating a fixed number of the vulnerable is clearly a better strategy for reducing disease than vaccinating the mixers. When the mixing is concentrated into a small core group (Figure S9 bottom right panel) the opposite is true, vaccination

personally, I would outline the analytical results for generic group size in the main text, with numerical results for $x=1$ (and the others as Supp Info)

of the mixers is vastly more effective in reducing disease.

It is unclear how exactly relative vulnerability and mixing should be modified here with changing
population sizes. In practice of course this is likely to be further modulated by their being more
than two groups, but rather a spectrum, and the relative balance in the most extreme groups for
vulnerability and mixing are likely to be important in determining optimal vaccination strategy.

Figure S1: Vaccine escape pressure

Each panel shows a contour plot vaccine escape pressure (as described in main text) plotted by proportion of vulnerable and mixers vaccinated (v_1 and v_2 respectively on horizontal and vertical axes). Note that for clarity in showing the shapes here, the contours and colours vary between panels (as the top left panel requires a much larger range than the others).

Different panels vary the effects of the vaccine: the rows step through $\theta_T = 1, 0.8, 0.6, 0.4$ and the columns step through $\theta_S = 1, 0.8, 0.6, 0.4$. Thus the top left corresponds to the vaccine having no transmission blocking effects at all, and stepping right and down increases transmission blocking through reduced susceptibility or infectivity. All other parameters are in Figure 1.

Figure S2: Sensitivity to contribution to vaccine escape

Vaccine pressure is now a linear combination of cases in vaccinated individuals (weight 100%) and unvaccinated individuals (weight varies). The weighting of the unvaccinated starts from zero and steps up by 2% in subsequent panels (left to right, top row then bottom row).

All parameters are as in Figure 1.

Figure S3: Summary outputs - with smaller G

This is as Figure 1, except $G = 5$.

Figure S4: Optimal for averting disease - with smaller G

This is as Figure 2, except $G = 5$

Figure S5: One dimensional paths to explore vaccine escape

This is as Figure 3, except $G = 5$

Figure S6: Vaccine escape pressure

This is as Figure S1, except $G = 5$

Figure S7: Sensitivity to vaccine escape pressure assumptions

This is as Figure S2, except $G = 5$

Figure S8: Changing proportions of vulnerable and mixers, fixed d and m

Different rows give sets of results different proportions of vulnerables and mixers corresponding to Figure 1 and the $\theta_S = \theta_I = 0.6$ panel of Figure 2 (and the contour colours also correspond to these plots in the main text). From top to bottom, vulnerables are 20%, 40%, 60%, 80% of the population and the rest are mixers.

Throughout, the relative vulnerability and mixing are kept fixed as the population proportion changes ($m = 2$ and $d = 10$).

Figure S9: Changing proportions of vulnerable and mixers, adjusting d and m

Different rows give sets of results different proportions of vulnerables and mixers corresponding to Figure 1 and the $\theta_S = \theta_I = 0.6$ panel of Figure 2 (and the contour colours also correspond to these plots in the main text). From top to bottom, vulnerables are 20%, 40%, 60%, 80% of the population and the rest are mixers.

The relative vulnerability and mixing are varied as the population proportion changes, to keep the excess constant (described in the text).

Appendix B

Prof. Julia Gog OBE

Department of Applied Mathematics and Theoretical Physics
Centre for Mathematical Sciences
Wilberforce Road,
Cambridge
CB3 0WA

**UNIVERSITY OF
CAMBRIDGE**

Email: jrg20@cam.ac.uk
Tel: +44(0)1223 760429

By upload as cover letter

24th June 2021

Dear Open Science,

We are resubmitting our manuscript entitled “Vaccine escape in a heterogeneous population: insights for SARS-CoV-2 from a simple model” to Royal Society Open Science, for consideration under the Science, Society and Policy section.

We are immensely grateful to both reviewers for thoughtful comments. We respond point by point below, and there are many consequent changes and additions to the manuscript.

On behalf of the authors, thank you again for your consideration of our work.

(detailed responses on subsequent pages)

Prof. Julia Gog, OBE
David N. Moore Fellow in Mathematics, Queens' College
Professor of Mathematical Biology, DAMTP, University of Cambridge
Member, JUNIPER consortium, <https://maths.org/juniper/>

Associate Editor Comments to Author (Professor Enrico Bertuzzo):

Associate Editor: 1

Comments to the Author:

Both reviewers found the manuscript interesting and sound but they also highlighted some areas of improvement, especially in the presentation. I welcome the authors to revise the manuscript according to these suggestions.

Thank you - our responses are interspersed below in this colour, and changes in the manuscript highlighted.

We are hugely grateful to both reviewers.

Reviewer comments to Author:

Reviewer: 1

Comments to the Author(s)

The manuscript by Dr. Gog and colleagues deals with the analysis of a SIR-like epidemiological model applied to the transmission of SARS-CoV-2. Using the model, the authors discuss several vaccination strategies for a population composed of subgroups characterized by different mixing and vulnerability patterns. The focus of the analysis, besides the derivation of standard epidemiological metrics such as the reproduction number and the incidence of infection, is on the possibility for vaccines to exert selection pressure on the virus, ultimately resulting in the emergence of mutations that may be able to escape the immune response triggered by the administration of the vaccine.

Needless to say, the topic is of extreme interest. The almost equation-free approach used by the authors may also serve well the purpose of widening the readership of an otherwise technical manuscript. The toy-like nature of the model seems to be better suited to seek general mechanisms rather than specific decision-making prescriptions. This point is effectively addressed in the manuscript and should not be seen, in my view, as a limitation of this study. The presented results seem sound, given the hypotheses laid out by the authors.

That being said, I have some technical comments that the authors may want to consider while revising their work:

The complexity of the model analyzed in the main text is kept to a minimum---and, I would argue, understandably so. Of the several simplifying hypotheses that have been introduced, one leaves me a bit perplexed, though: namely, that the two groups have the same relative abundance within the population. Besides the obvious unlikelihood of such numerical coincidence, I wonder whether this choice could perhaps lead to an underestimation (not quite by the authors, rather by some readers) of possible asymmetries in the transmission process and in the definition of epidemiological patterns. I am especially referring to the analytical treatment, where the 'vulnerable'-to-'mixer' ratio is nowhere to be found, exactly because of this strong 1-to-1 hypothesis. However, this ratio influences several of the results presented in this work, as acknowledged (and even shown) by the authors. I would suggest removing this 1-to-1 hypothesis from the main text while keeping all the other simplifications in place. Numerically, I would not change anything, meaning that the main text could still just account for the case $\epsilon=1$ (borrowing from the extended model presented in the main text).

The construction of two equally sized groups was a decision we took, but on reflection the rationale for doing this could be made clearer.

Longer version of the thinking: in early rounds of model development, we considered three population segments - the bulk of the population that were neutral in vulnerability and mixing, then a minority with higher vulnerability and a minority with higher mixing. This is clearly still an extreme

caricature of reality. However, this means we still require four parameters (vulnerability and mixing strengths, and the two population sizes). We were seeking the simplest approach to capture the relevant heterogeneity that illustrates the key effects. The neutral population wasn't needed, only the mixers and the vulnerable.

It certainly is important to explore splits other than 50:50, hence it is included in the Supplementary Information. One could argue that the vulnerable are a small group if we take them to represent say the over 75s, or to correspond to the clinically extremely vulnerable of the UK vaccination phases. Equally, we could argue that the mixers are a small group, say adults aged 18-25. Instead, taking the simplest heterogeneity (equal split) seems parsimonious. Reality is of course something more like a joint distribution of vulnerability and mixing, but rather than seeing the equal split as a coincidence, it can be taken as one way of abstracting this distribution. In terms of age, this could be those under and over 40 (roughly the UK median age).

It would have been nice to find a way to include the varying split proportions in the main text, but then, for example, the already-complex figure 1 has to become something like either figure S8 or S9, and the expressions in 2.3 become cumbersome. In addition, there is the decision on how to vary the strength of mixing and vulnerability with the proportions changing (the difference between S8 and S9). We think varying this many properties would detract too far from clarity for most readers. For future work, rather than exploring this split further, we would instead recommend considering more nuanced distributions more closely reflecting reality, but that is the start of a further study.

We have added a bit more in the methods section on "population heterogeneity", expanded why equal sizes is reasonable for illustration in the main text, and also expanded the methods for more general population structures.

I am not against the 'direct calculation' approach chosen by the authors for the definition of the next-generation matrix. However, equation (2) needs to be better framed and more explicitly explained to make sure that readers can easily follow. For instance, I believe that at least some future readers might be left somehow dumbfounded by the fact that the fraction of vaccinated infectious people does not appear in the last two entries of the first row of the next-generation matrix (similar remarks apply to other entries as well). A similar observation holds also for equations (3) and (4), which are introduced basically with no prior methodological background. I believe that in all these cases the authors would do the less mathematically-versed readers a solid if they could expand just a bit the explanation of these technical aspects of their work.

This, and the comments here on the manuscript are extremely useful feedback, thank you. There is nothing very deep going on in the methods, but certainly we could make things clearer for the reader, though this does require expanding a bit.

For understanding how (2) appears as it does, perhaps the key information is that the next generation matrix has an inbuilt asymmetry: if K is the NGM (which is proportional to M) then K_{ij} is the number of infections in group i that would be caused by ONE infected in group j . Hence this process for splitting a population group (without changing underlying mixing): duplicate the column but split the row. This is what is going on with the fraction vaccinated going by row only (and similarly the group sizes only appearing in the first vector in (14)).

We have reworked the text in this section to walk the reader through the key steps, including building the 4x4 matrix. This has lengthened this section, but we believe from this reviewer's comments that these details are worth including.

Some epidemiological terms need to be better defined. For instance, I cannot fully understand what do the authors mean when they say that in their model "incidence $I(t)$ is exponential, with growth rate λ " (p.4, l.39). Now, if they have in mind a model like $dI/dt = \lambda I$, then I guess that $I(t)$ would be the cumulative incidence at time t ; if so, I do not get where the integral in equation (5) comes from. Some further explanation seems to be warranted here. The same goes for the term "prevalence", which seems to be used naively (in both the abstract and the summary).

We think the confusion here might be caused by our use of $I(t)$ for incidence (the number of new cases per day), when usually the variable I in an SIR model represents something more akin to prevalence (the total number who are infected/infectious on each day).

The final expression for $F(R)$ is as intended, but to mitigate the potential for confusion, we will change incidence to being denoted by y . The use of the term prevalence in the abstract and elsewhere is appropriate.

(In response to green handwritten comments here - If $I(t)$ is intended to be the prevalence, then $I' = \lambda I$ is not right either: Loss from recovery would also need to be included, but note we are not assuming any time to recovery distribution, as this is not needed in our direct formulation. If $I(t)$ in the handwritten comments is meant to be incidence, then this is equivalent to what we have, but it needs to be cumulative incidence *since* $t=0$ so subtract off I_0 , then the result would match ours. But, the integral of incidence seems to be the easiest route here.)

Selection pressure and vaccine escape are admittedly described quite naively in this work. I do not have objections to simplicity if put in perspective (as the authors do). However, I wonder whether it could be possible to translate the current definition of vaccine escape, which is not completely obvious to get dimensionally, to something like the probability of vaccine escape. I believe that this could be done quite easily (although perhaps at the expense of one additional parameter) if one defines $\text{Prob}(\text{vaccine escape}) = 1 - \text{Prob}(\sim \text{vaccine escape}) = 1 - (1-p)^{C \cdot P}$, with p being the probability of vaccine escape within a single host.

Our approach of looking at the number of cases in vaccinated individuals is essentially to give the exponent (or something proportional to the hazard) in any expression of this sort. The parameter given as p by the referee (probability of escape per case) is extremely problematic to estimate, with significant heterogeneity between different infected hosts (though we nonetheless explore it a bit here: [https://www.thelancet.com/journals/laninf/article/PIIS1473-3099\(21\)00202-4/fulltext](https://www.thelancet.com/journals/laninf/article/PIIS1473-3099(21)00202-4/fulltext)).

On balance, in the context of this work, we think it justified to keep the measure of vaccine escape pressure as something proportional to the number of cases (hazard). We do fully agree that the evolutionary aspects here have been addressed in quite naive terms - the price is some realism, but we gain tractability, transparency and generalisability to multiple vaccination scenarios.

Following up on the previous point, the authors assume (in the main text) that only do infections in vaccinated people contribute to the risk of vaccine escape. However, they acknowledge that the situation is much more complicated in reality, and even relax their hypothesis (in the supplements) by accounting for the role possibly played by infection in unvaccinated people. As a matter of fact, every infection gives the virus new chances to evolve, by genetic drift if not by selection. With viral transmission still rampant and vaccine rollout still slow in many countries, understanding what mechanism contributes the most to evolutionary dynamics is of course challenging (leaving aside competition dynamics, which would require a more complex modeling framework). That is why it would seem important to me to include at least part of the section about the sensitivity analysis of vaccine escape results, along with Figure S2, in the main text.

Understanding the limitations of our approach and the sensitivities is important. For the point about unvaccinated people contributing towards escape pressure (rather than purely the vaccinated people as we've assumed in the main text), this does not require any additional methods, and also can more or less be read off from our results (just linearly interpolate from V to C - or bottom right to top right of Figure 1). Given that this extension does not do anything unexpected or add any new insights (and it is really only one small step of adding detail, there are so many other simplifications that we have made that could arguably be considered before or with this), it seems right to leave it in supplementary material. However, given the comments of the reviewer, which may also come to mind for other readers when reading our manuscript, we have extended the relevant results section with these points.

The manuscript is generally well written and quite easy to follow. However, there exist several instances where writing could be further improved for clarity. I am attaching a copy of the manuscript file with some minor remarks and suggestions marked in green (plus some notes of mine which have been translated into the comments above).

Absolutely amazing! Please pass on our gratitude to the referee for taking such time and thought here. We have made nearly all of the changes exactly as suggested. For the remaining few, we have made slightly different changes in response as we could see what the issue was. For the points that are already mentioned above, it was very helpful to understand where exactly the confusion starts. We are sure these suggestions from this referee will have helped to improve the clarity of the manuscript.

Reviewer: 2

Comments to the Author(s)

The manuscript "Vaccine escape in a heterogeneous population: insights for SARS-CoV-2 from a simple model" by Gog et al. analyses a simple model for vaccination in a heterogeneous population, to infer some general principles, that may be useful for designing actual vaccination strategies.

In a stylized population consisting of two groups, one with a higher contact rate, the other one subject to more serious complication if infected, the authors study in which group it is more convenient allocating limited vaccine resources, according to different criteria.

The model is simple enough that analytical formulae can be obtained and computed to answer the question. The answer depends of course on parameter values and on the criterion used; the authors conclude anyway that "in the majority of the parameter space explored, vaccinating the mixers is more effective than vaccinating the vulnerable to reduce the total amount of disease". This result, valid as long as vaccines are able to limit, at least partially, the transmission of the infection and there is a significant difference in contact rates between the two groups, is in line with the general epidemiological theory. I must however remark that, if we are thinking of COVID-19 and the groups represent different younger and older age classes, the value of the parameter d should be around 1,000 (see, e.g. O'Driscoll et al, 2021) rather than in the range 1-10, and this would make quite a difference. Possibly this is one of the reasons for the different result obtained in [44], beyond the ones offered by the authors. I think that the authors should at least acknowledge the issue.

Putting a scale on d is very difficult here. In part this is because of the crude population split into only two groups. For example, if we take the population median age to be around 40 and estimate the population weighted IFR from O'Driscoll et al Figure 2a for the younger and older half of the population - this looks to us to be more like 100 than 1000. Further, taking "severity" as hospitalisations would probably give a lower d than deaths.

However, in any case, our results really are not very sensitive to d once it is soundly over 1. In essence, going from $d=10$ to $d=1000$ means weighting the mixer cases' contribution to D as 0.1 or 0.001. We have added some further explanation to the parameter estimation section on this, and include a new section in the Supplementary Information with versions of the key figures 1 and 2 with $d=1000$ for illustration. We thank the reviewer for directing us to think again on this.

The more novel part of the article concerns the effect of vaccination policy on the probability of vaccine escape. While the model is very simple and the results are difficult to interpret in terms of actual policies, it is important bringing the point to both modellers and public health authorities, and the general principle (intermediate vaccination rates maximize the risk) appears to be robust.

I think that the manuscript is interesting and worthwhile. The authors recognize the limitation of the model used, and they discuss with competence whether their results are expected to be robust to model details.

In the Supplementary Material the authors show the effect of some changes in the model or in the parameter values used. I would have been interested in seeing the effect of at least two other modifications:

- the authors always assume proportional mixing among the two groups. What if mixing is to some degree assortative?

Another good question. Unfortunately it would break our analytic approach (matrices could not be written as outer products in general). The form we have at the moment is the most indiscriminate mixing - our mixers have higher mixing rates but they just mix with whoever else is out there mixing rather than an additional preference for other mixers. Our intuition is that anything to make things more assortative than they are will have the effect of just further boosting the importance of mixers in shaping R . Hence our core insights (the value of using vaccination to lower R , the highest risk if targeting vaccination to the vulnerable) will, if anything, be emphasized further. Our current assumption is probably conservative with respect to our results.

However, it is not clear the strength of this effect, and also how far this intuitive prediction could be pushed. For example with more age classes and population classes, it might matter which groups are core mixing and how they are connected to the most vulnerable groups.

We don't think we can offer further mathematical work here without moving to a different approach, at which point it would make sense instead to use more realistic age and population mixing. While we are not comfortable adding any further speculation to the manuscript on this topic, we do think that our results are very likely to be robust to further work in this direction, for the reasons above.

- the model assumes that some part of the population is vaccinated at $t=0$, and then the epidemic proceeds exponentially according to the resulting parameter values. Would the picture be different if vaccinations occur dynamically? Namely, they occur at some prescribed rate during the time period analysed. I understand that the problem is much more complex, as there would be no simple formula to evaluate the output, and simulations would be required. Furthermore, the model could become more complex, as one may think that public health authorities relax NPIs as a larger fraction of the population becomes vaccinated, bringing economic issues in the optimization, as already suggested by the authors at page 15. Still, I think it is an issue that is worth being analysed in as simple a context as possible.

If the authors find the time to briefly analyse these issues, I think it would be an interesting addition to the manuscript, but this is only a suggestion.

These are all excellent points. Bringing in some of the dynamic problems should also entail changes in vaccine efficacy at the individual-host level over time (rather than assuming that vaccines are effective immediately following vaccination, and that immunity does not wane) and dosing regimes. This does move things firmly beyond the capacity of the simple dominant eigenvalue approach and into the domain of more detailed simulations. We agree that this is worth doing, and that the links with economic considerations are also important.

However, we believe that this has the makings of a full research programme, requiring significant extensions to the simple analytic framework presented here. We hope that the insights and principles illustrated in this manuscript stand alone without exploring these further directions, which we intend to investigate in future with a more complex simulation model. We also hope that by sharing our work and discussion in this paper, others may also be encouraged to explore this important and interesting area further.

Reference cited

O'Driscoll, M., Ribeiro Dos Santos, G., Wang, L., Cummings, D. A. T., Azman, A. S., Paireau, J., Fontanet, A., Cauchemez, S., & Salje, H. (2021). Age-specific mortality and immunity patterns of SARS-CoV-2. *Nature*, 590(7844), 140–145. <https://doi.org/10.1038/s41586-020-2918-0>